# Nettle-Leaf Extract Derived ZnO/CuO Nanoparticle-Biopolymer-Based Antioxidant and Antimicrobial Nanocomposite Packaging Films and Their Impact on Extending the Post-Harvest Shelf Life of Guava Fruit

**DOI:** 10.3390/biom11020224

**Published:** 2021-02-05

**Authors:** Anu Kalia, Manpreet Kaur, Ashwag Shami, Sukhjit Kaur Jawandha, Mousa A. Alghuthaymi, Anirudh Thakur, Kamel A. Abd-Elsalam

**Affiliations:** 1Electron Microscopy and Nanoscience Laboratory, Department of Soil Science, Punjab Agricultural University, Ludhiana 141004, Punjab, India; 2Department of Fruit Science, Punjab Agricultural University, Ludhiana 141004, Punjab, India; manpreet2126@gmail.com (M.K.); skjawandha@pau.edu (S.K.J.); 3Biology Department, College of Sciences, Princess Nourah bint Abdulrahman University, Riyadh 11617, Saudi Arabia; AYShami@pnu.edu.sa; 4Biology Department, Science and Humanities College, Shaqra University, Alquwayiyah 11971, Saudi Arabia; 5Plant Pathology Research Institute, Agricultural Research Center (ARC), 12619 Giza, Egypt

**Keywords:** anti-microbial, chitosan, guava, nanocomposite, perishable fruits, phyto-synthesis, storage shelf-life

## Abstract

Green synthesized metal oxide nanoparticles (NPs) have prominent applications in antimicrobial packaging systems. Here we have attempted for the fabrication of chitosan-based nanocomposite film containing *Urtica dioica* leaf extract derived copper oxide (CuO) and zinc oxide (ZnO) NPs for shelf-life extension of the packaged guava fruits. Electron microscopy and spectroscopy analysis of the CuO and ZnO NPs exhibited nano-scale size, spherical morphologies, and negative ζ-potential values. The NPs possessed appreciable antioxidant and antimicrobial activity (AMA) in order of CuO NPs > ZnO NPs > nettle extract. Therefore, this work establishes for the first time the successful synthesis of CuO NPs and compares its antimicrobial and antioxidant properties with ZnO NPs. On incorporation in chitosan, the polymer nanocomposite films were developed by solvent casting technique. The developed films were transparent, had low antioxidant but substantial AMA. The NP supplementation improved the film characteristics as evident from the decrease in moisture content, water holding capacity, and solubility of the films. The nanocomposite films improved the quality attributes and shelf life of guava fruits by one week on packaging and storage compared to unpackaged control fruits. Therefore, this study demonstrates the higher antimicrobial potential of the nettle leaf extract derived CuO/ZnO NPs for development of antimicrobial nanocomposite films as a promising packaging solution for enhancing the shelf life of various perishable fruits.

## 1. Introduction

Guava (*Psidium guajava* L.) is a climacteric fruit grown in the sub-tropical and tropical countries of the world. In India, guava is grown on 265 thousand ha of land with an annual production of 4.05 million tons [1]. Guava fruits are rich in dietary fiber, pectin (0.5 to 1.8%), vitamins such as vitamin B1, B2, B3, C, minerals, carotene, calcium, and iron [2]. Its consumption reduces serum cholesterol and triglycerides levels besides increasing the high-density lipoprotein [3]. However, it is highly perishable with a shelf life of approximately one week at ambient temperature and ~two weeks at 6 to 8 °C [4]. Hence, it suffers from substantial post-harvest losses during storage and transportation.

Many synthetic and non-degradable polymers such as polyvinylchloride, polyvinylpyrrolidone, low- and high-density polyethylene have been used for developing packaging materials to address the spoilage losses incurred during storage and transportation of the guava fruits. Inappropriate disposal of these packaging materials after use leads to chronic release of hazardous material(s) in the environment that can have serious ecosystem deterring implications. Therefore, the escalating waste disposal issues associated with the use of synthetic packaging materials are paving research endeavors towards the use of possible alternative environment-friendly biodegradable polymers.

One plausible alternative is the development and use of degradable polymer packaging films derived from polyvinyl alcohol [5], chitosan [6], starch [7], and polymer blend substrates [8,9]. However, the degradable polymeric films exhibit inferior mechanical, thermal, and gas barrier properties. Therefore, supplementation with nano-filler agents such as nanoparticles (NPs) to form polymer nano-composites can be a useful alternative to address the limitations of the polymeric films. The nanocomposite films exhibit better thermal stability besides additional characteristics such as antimicrobial potential against bacterial and fungal pathogens can be achieved [10].

Among the natural biodegradable polymers, chitosan—deacetylated chitin derivative (polysaccharide of N-acetyl D-glucosamine and D-glucosamine units) possesses good mechanical and antimicrobial properties [11]. Chitosan act as a promising biopolymer for the use in food preservation and packaging applications since it is easily available, biodegradable, non-toxic, can bind to ions (particularly exhibits affinity for Cu^2+^ and Zn^2+^ ions) and exhibits solubility in acidic conditions [12]. The incorporation of natural extracts in chitosan can further improve the antimicrobial potential of the developed films [13,14]. The chitosan-based nanofilms are generally supplemented with the nano-filler agent(s) to improve their properties. The most common nano-filler agents used for the synthesis of antimicrobial polymer nanocomposite films are the metal/metal oxide NPs. Such NPs supplementations have been reported for the development of novel nano-composite packaging films by incorporation of polyethylene (PE) with nanoparticles of silver, titanium dioxide, and swelling clay such as montmorillonite [15]. These films exhibit improved gas barrier properties, water vapor permeability, and mechanical strength. Therefore, for the fabrication of natural polymer-based nanocomposite films low cost and ecologically safe nanoparticles are required.

Metal oxide NPs generated through chemical and physical techniques have naïve external surface and therefore, these NPs exhibit aberrant features such as high-water repellence making them undesirable for biological applications. However, the green synthesized NPs involve natural reducing and biotemplating molecules which also play role in priming or functionalization of the NPs. The biomolecules which reduce metal salts and lead to nucleation of metal/metal oxide NPs can be obtained from a cell or cell-free extracts of diverse living cells including prokaryotic bacteria [16], eukaryotic algae, fungi [17], plant [18] and animal cells. Among these biological entities, the use of plant extracts including the herbs of biomedical importance provides a simpler and robust method for facile and scaled-up synthesis of metal/metal oxide NPs [19].

Stinging nettle (*Urtica dioica* L.) is a weed that exhibits rapid growth and soil coverage and has been known to improve soils over-fertilized with inorganic fertilizers such as nitrogenous and phosphatic fertilizers or soils contaminated with heavy metals [20]. The phytoremediation properties of stinging nettles can be biochemically ascribed to the abundant occurrence of diverse bioactive molecules including small primary/secondary metabolites (amino acids, carotenoids, chlorophyll, vitamins) and other compounds (fatty acids, hydrocinnamic acid, polyphenolic compounds, polysaccharides, sterols, tannins, isolectins, iron, and terpenoids) [21]. The phenolic content in nettles (129 mg Gallic acid equivalent (GAE) per gram of nettle powder) has been observed to be richer in certain polyphenol compounds such that it can be two-time higher than the cranberry juice [22]. These phytochemicals act as a reducing agent for metal/metal oxide salts leading to the generation of NPs [23]. The synthesized NPs get coated with the nettle leaf phytochemicals and therefore, are anticipated to possess improved antimicrobial potential.

The present investigation aimed for the fabrication of chitosan-based antimicrobial packaging films containing stinging nettle leaf extract enabled phyto-synthesized CuO and ZnO NPs. The primary hypothesis for the use of copper and zinc oxide NPs from the nettle leaves was based on the potent antioxidant, anti-bacterial and anti-fungal activity of the copper and zinc salts against the predominant spoilage and opportunistic pathogens of guava fruit. Further, the abundance of phenolic and other biomolecules of antimicrobial potential in nettle leaves was focused to be utilized for the generation of ZnO and CuO NPs which may exhibit improved or augmented anti-bacterial and anti-fungal properties. The use of ZnO and CuO NPs were expected to improve the shelf life of guava by decreasing the spoilage causing microbes through nano-scale phenomena. The novelty of this report is based on the use of leaf extract of stinging nettle for the synthesis of CuO and ZnO NPs and their incorporation in chitosan biopolymer for the development of nanocomposite packaging films for guava fruits.

## 2. Materials and Methods

### 2.1. Materials

#### 2.1.1. Chemicals

Analytical grade chemicals were utilized for carrying out the experiments. Chitosan (Deacetylation degree ≥ 90%, water 10%, ash 2%, medium (190,000 to 300,000 g mol^−1^) molecular weight), zinc acetate, copper sulfate, acetic acid, 1,1-Diphenyl-2-picrylhydrazyl (DPPH), 2,2′-azino-bis(3-ethylbenzothiazoline-6-sulfonic acid (ABTS), methanol, and chloramphenicol antibiotic were purchased from Hi-Media Laboratory Chemicals Pvt. Ltd., Mumbai, India. Commercial ZnO NPs were purchased from Sisco Research Laboratory Pvt. Ltd., Mumbai, India.

#### 2.1.2. Microbial Cultures

Four human pathogenic microorganisms namely *Enterobacter cloacae* MTCC 509, *Salmonella typhii, Staphylococcus aureus*, and *Campylobacter jejuni* were used for the antimicrobial studies. The culture *E. cloacae* MTCC 509 was procured from Microbial Type culture collection, IMTECH, Chandigarh, and the remaining three cultures were procured from the Department of Microbiology, Punjab Agricultural University, Ludhiana, India.

#### 2.1.3. Guava Fruits

The fruits of guava cv. Allahabad Sufeda of appropriate maturity or age were collected from Fruit Research Farm, Department of Fruit Science, PAU, Ludhiana, India located ~two km from the laboratory.

### 2.2. Methods

#### 2.2.1. Biosynthesis of Nanoparticles

Leaf extract (10% *w/v*) of sting nettle (*Urtica dioica* L.) was prepared by boiling nettle leaves in deionized water followed by filtration [23]. Briefly, the dried nettle leaves (10 g) were added to 100 mL of deionized water and boiled for 30 minutes on a heating plate/mantle under reflux conditions. The extract was filtered through filter paper (Whatman qualitative filter paper, grade 42, USA) to remove the leaf debris or other particles. The filtrate was collected, cooled, and stored under refrigerated conditions for any further use. The copper sulfate and zinc acetate (0.01 M) solutions were also prepared in deionized water. The synthesis of NPs involved the drop-wise addition of the nettle extract (10% *v/v*) in the bulk salt solutions with constant stirring (800 rpm) on a magnetic stirrer at room temperature for 30 min. The pH of the final solutions after addition of the nettle leaf extract was maintained at neutral (7.0) and alkaline (9.0) for CuO and ZnO NPs respectively. The initial assertion for the formation of CuO and ZnO NPs was obtained by visualizing the color change of the solution which turned green and milky white respectively. The color change was considered indicative of the formation of NPs since the bulk salts were reduced by the biomolecules present in the nettle extract.

#### 2.2.2. Nanoparticle Characterization

The nettle leaf extract derived NPs were characterized through spectroscopy and microscopy techniques. The UV-Vis spectra were obtained for the wavelengths ranging from 200 to 800 nm on UV-Vis Spectrometer (Elico SL-218, Hyderabad, Telangana, India). For elucidation of the element composition of NPs, the analysis was performed with an energy dispersive X-ray spectroscope (EDS, Thermo Noran, Waltham, MA, USA) configured with the Scanning electron microscope (SEM, Hitachi s-3400N, Minato-Ku, Tokyo, Japan). The FT-IR spectra of the biosynthesized NPs were recorded from 4000 to 400 cm^−1^ wavenumbers on FT-IR spectroscope (Thermo Nicolet 6700 NXT, Thermo Fischer Scientific, Waltham, MA, USA). Transmission electron microscopy (TEM, Hitachi H-7650, Minato-Ku, Tokyo, Japan) studies of the NPs was performed to obtain the average size and shape of the NPs. A drop (10 µL) of the NP sol obtained after sonication was placed on copper grids (carbon-coated, 200 mesh size, Tedpella, Redding, CA, USA). The TEM images were captured in high contrast (HC) imaging mode. Dynamic light scattering spectroscopy (DLS) and zeta potential (Beckman Coulter Delsa Nano, Indianapolis, IN, USA) studies were performed for aqueous suspension of nanoparticles after dilution and bath sonication treatment.

#### 2.2.3. 2,2-Diphenyl-1-Picryl-Hydrazyl (DPPH) Radical Scavenging Activity (RSA)

The antioxidant potential of the nettle extract and synthesized NPs was determined by the DPPH RSA method [24]. Nettle extract and biosynthesized NPs (0.05 mL) were incorporated in 1.95 mL of methanolic DPPH (0.1 mM) solution and mixed properly. The mixture was incubated for 45 min at 25 ± 2 °C or room temperature in dark condition and later absorbance was read at 515 nm. The DPPH RSA potential of the nettle extract, CuO and ZnO NPs was compared with the values obtained for the butylated hydroxytoluene (BHT) as standard.
(1)$$DPPH\;radical\;scavenging\;activity\;\left(\%\right)=\left[\frac{Absorbance\;\left(DPPH\;solution\right)-Absorbance\;\left(Test\;sample\right)}{Absorbance\;\left(DPPH\;solution\right)}\right]\times 100$$

#### 2.2.4. Antimicrobial Activity (AMA) of Synthesized Nanoparticles

Biologically synthesized ZnO and CuO NPs were tested against four pathogenic cultures viz., *Enterobacter cloacae* MTCC 509, *Salmonella typhii*, *Staphylococcus aureus*, and *Campylobacter jejuni*. The AMA was determined through a modified agar well technique. Lawn cultures of the test microorganisms were prepared using freshly grown cultures in the log phase of growth. Later, wells (5 mm diameter) were prepared aseptically using a cork borer, and the two types of NPs (@ 25 µL) were added separately. The commercial formulation of NPs and Chloramphenicol antibiotic (@ 30 mcg) was kept as positive controls while the nettle extract was kept as a negative control. The inoculated plates were placed in a BOD incubator at 37 °C for one day (24 h) and thereafter, the inhibition zone (ZOI, mm) thus formed was recorded.

#### 2.2.5. Fabrication of Chitosan Nanocomposite Films

Chitosan (1% *w/v*) was dissolved in acetic acid (1% *v/v*) with constant stirring at 40 °C for 2 h. The synthesized NPs (@ 25 mg L^−1^) was added followed by stirring for another 2 h. The chitosan-NP solution was bath sonicated (Toshcon, Ajmer, Rajasthan, India) for half an hour to improve the dispersion of the supplemented NPs in the chitosan solution. The chitosan-NP composite films were developed by casting technique which involved pouring of sonicated formulations in teflon coated baking trays. The trays were placed undisturbed on a levelled benchtop at 25 ± 2 °C and were allowed to set for 12 h. Once set, the films were then placed in a hot air oven (REMI RDHO, REMI Electrotechnik Ltd., Thane, Maharashtra, India) at 60 °C. The dried films were peeled off and stored in desiccators at ambient temperature with 60% relative humidity for further use.

#### 2.2.6. Antimicrobial Activity of Chitosan Nanocomposite Films

The film AMA was determined by a modified disk diffusion assay [25] against the four pathogenic cultures as described in the AMA of the synthesized NPs section. Film disks (5 mm) were cut and placed on the surface of bacteria inoculated nutrient agar medium followed by incubation at 37 °C in a BOD incubator for 24 h. The ZOI around the disks was recorded after 24 h as an indicator of the lysis or killing of the test bacteria. The standard antibiotic discs were used as positive controls.

#### 2.2.7. Physical Characterization of Films

Physical properties including thickness, color, transparency, surface morphology, and elemental composition of the prepared films were studied.

##### Film Thickness

The thickness of the prepared films was determined with a digital micrometer (Mitutoyo, Japan). Random measurements were performed on different regions (at least 10 places) on the prepared films.

##### Surface Color Measurement of Films

The film color was measured as per the CIE (L *, a * and b * value) model on a Hunter Lab Colour difference meter (Color Flex^®^, Reston, VA, USA) from three regions of the film. The range for the three variables of the CIE system include L * = 0 (black) to 100 (white), a * = redness (+a) or greenness (−a) and b * = yellowness (+b) or blueness (−b) colour. The values were obtained by using the following equation
Chroma (C) = (a^2^ + b^2^)½, hue angle (ho) = tan−1 b/a(2)

##### Transparency

The film transparency studies were performed in a UV-Vis spectrometer using a rectangular section (7 × 30 mm) of the developed films with absorbance peaks recorded at 600 nm as per the method [26] and equation
(3)$$Transparency=-\log\left(\frac{T600}{Th}\right)$$
where, *T*600 = transmittance at 600 nm and ‘*Th*’ = film thickness (mm).

##### Film Moisture

The moisture content (MC) of the prepared films was determined by taking the weight (*W*1) of small pieces of films (Size- 3 × 3 cm^2^) which were placed in an RH maintained enclosure. These sections of the films were then dried in a hot air oven at 105 °C for one day (24 h). The weight recorded after drying (*W*2) was taken as the dry weight of the sample. The percentage of *MC* was calculated as
(4)$$MC\;\left(\%\right)=\left[\frac{W1-W2}{W1}\right]\times 100$$

##### Water Holding Capacity

For determining the percent water holding capacity (*WHC*), film sections of area 2.5 × 2.5 cm^2^ were weighed (initial weight, *Wi*). Then the films were immersed for 2 min in distilled water and were removed followed by removing excess water by filter paper. After this, the final weight (*Wf*) was measured.
(5)$$WHC\;\left(\%\right)=\left[\frac{Wi-Wf}{Wi}\right]\times 100$$

##### Film Solubility

For measuring the film solubility (*FS*), small pieces of films (Size-3 × 3 cm^2^) were cut and dried for 2 h in a hot air oven at 60 °C. After drying, the initial weight of these film pieces was determined (*Wi*). Then these sections of films were immersed in distilled water (30 mL) for one day (24 h) with occasional shaking. Afterwards, the remnant film sections were collected and dried at 100 °C in an oven for 24 h. The final weight (*Wf*) of the film sections was recorded after drying.

The formula for percent film solubility (*FS*),
(6)$$FS\;\left(\%\right)=\left[\frac{Wi-Wf}{Wi}\right]\times 100$$

#### 2.2.8. Chemical Structure Characterization of Prepared Films

##### SEM and SEM-EDS Analysis of the Films

The surface morphology of the nanocomposite films was viewed on SEM (operated at 15 kV acceleration voltage in SE imaging mode). The elemental composition of the surface of the film samples was determined on EDS configured to SEM and the data was presented as percent atom and weight.

##### Antioxidant Properties of Prepared Chitosan Nanocomposite Films

The chitosan nanocomposite films (50 mg) were placed in methanol and homogenized properly in an autoclaved pestle and mortar kept in crushed ice for 10 min. The crushed sample was collected and centrifuged at 15,000× *g* for 15 min in a refrigerated centrifuge (Remi CPR-30 Plus, Mumbai, Maharashtra, India) at 5 °C. The pellets were discarded, and the supernatant was then filtered through filter paper (Whatman qualitative filter paper, grade 1, Buckinghamshire, United Kingdom). The filtrate obtained was assayed for DPPH radical scavenging activity in triplicate [24]. The ABTS (2,2′-azino-bis(3-ethylbenzothiazoline-6-sulfonic acid) scavenging activity was evaluated as per the method of Zheng et al. with some modification [27].

#### 2.2.9. Physiological, Biochemical, and Microbiological Characterization of Guava Fruits

Guava fruits were sorted to remove any unripe and damaged fruits. Around 3 kg of fruits of uniform size and color were selected and wrapped in each NP-incorporated and control chitosan films. After wrapping, 6 fruit each were packed in corrugated boxes (dimensions 21 × 15 × 13.5 cm) and stored at 6 °C and 90–95% RH in walk-in cold rooms in Postharvest Laboratory, Department of Fruit Science, PAU, Ludhiana, India. While around 4.5 kg of fruits packed in the CFB boxes were kept unwrapped which were considered absolute control. The observations were taken at 7 days interval up to 3 weeks (21 days) of storage.

##### Weight Loss

After packaging, the fruits were weighed at specific time intervals using the following formula to determine the weight loss.
(7)$$Weight\;loss\;\left(\%\right)=\left[\frac{Initial\;weight-Final\;weight}{Initial\;weight}\right]\times 100$$

##### Fruit Firmness

The fruit firmness was determined with a penetrometer (FT-327, Norfolk, VA, USA) having a stainless-steel probe (8 mm) fitted on manual test stand for uniform application of force. About one cm^2^ peel of the fruit was removed from equatorial planes with the help of a peeler. The firmness was measured after the removal of a thin slice of peel from opposite equatorial points on the fruit surface. The force applied to plunge the 8 mm stainless steel probe into the peeled guava flesh on the manual test stand was recorded and expressed in Newtons. Three readings from different regions (three) per fruit were recorded and expressed as newton (N).

##### Soluble Solids

Fresh pulp juice was extracted from guava fruits for the estimation of the soluble solid content (SSC) with an infrared digital refractometer (Atago PAL 1 3810, Minato-ku, Tokyo, Japan). The results were expressed in °Brix values.

##### Titratable Acidity

Titratable acidity (TA) of the homogenized fruit pulp was determined by dissolving a known amount of fruit pulp juice in a known volume of distilled water. The solution thus obtained was titrated against sodium hydroxide solution (0.1 N) to light pink endpoint (pH 8.1) using phenolphthalein indicator [28]. The TA was expressed on a fresh weight basis as % citric acid.
(8)$$TA\;\left(\%\right)=\left[\frac{0.0064\times Titre\times Volume\;made}{Volume\;of\;fruit\;pulp\;juice\times Volume\;of\;aliquot\;used\;for\;titration}\right]\times 100$$

##### Sensory Quality

The samples were judged for organoleptic sensory rating by a panel of five individuals. The fruit was evaluated based on physical (appearance, color, freshness) and organoleptic parameters (texture, aroma, and taste). The overall acceptance was determined as per the 9-point hedonic scale [29].

##### Spoilage of Guava Fruits

The spoilage of fruits during storage was expressed as a percentage and enumerated as the number of fruit spoiled at specific storage interval.
(9)$$Spoilage\;\left(\%\right)=\left[\frac{Number\;of\;fruit\;spoiled}{Total\;number\;of\;fruits}\right]\times 100$$

Microbial fruit spoilage was estimated by dilution spread plating of wash water of the guava fruit surface on four different agar-based media specific for enumeration of viable cell count of total aerobic bacteria (TAB), single-cell yeasts, multicellular filamentous molds, and fecal bacteria belonging to Enterobacteriaceae family [30].

## 3. Results

### 3.1. Nanoparticle Characterization

The nettle leaf extract derived nanoparticles were characterized through various spectroscopy and microscopy techniques. UV-Vis spectroscopy analysis showed the occurrence of characteristic surface plasma resonance peaks (SPR) due to the formation of NPs. Both CuO and ZnO nanoparticles exhibited specific UV absorbance peaks at 320 nm and 330 nm wavelength respectively (Figure 1A). However, the UV-Vis spectra of ZnO NPs also exhibited shoulder peaks at 390 and 420 nm possibly indicating the formation of agglomerates of a bigger size.

The EDS analysis confirmed the occurrence of Cu (40.32 wt %) and Zn (40.14 wt %) elements on the NP surface based on the appearance of specific peaks (energy keV) of Kα (Cu-8.040, Zn-8.630) and Lα (Cu-0.930, Zn-1.012) X-ray lines in EDS spectra (Figure 1E). The atom% values of Cu and oxygen was 16.35 and 46.98 respectively (ratio- 1:2.87). While for the Zn and O elements the atom% was 15.56 and 45.74 (ratio- 1:2.93). The relative ratio of the Cu:O and Zn:O as calculated through % atom contents indicated the formation of CuO and ZnONPs as the ratio suggested the co-existence of the oxide and peroxide form of both the elements i.e., CuO/CuO_2_ and ZnO/ZnO_2_ respectively. Further, the higher oxygen atom% also indicates towards the contribution of organic compounds derived from the nettle leaf extracts to the oxygen signals in the EDS spectra. Therefore, the oxygen atom% are substantially higher than anticipated to derive the empirical formula for CuO/CuO_2_ and ZnO/ZnO_2_ NPs.

The FT-IR spectrum of the nettle leaf extract showed a broad vibration peak at 3324.68 cm^−1^ corresponding to stretching of hydroxyl (O-H) functional group. The peak for the C-H stretching vibration (2970 cm^−1^) was also recorded (Figure 1B). The stretching vibration peak at 2917.77 cm^−1^ indicated N-H stretching and the presence of an amine group. A peak at 1636.30 cm^−1^ depicting carbonyl group (C=O bond) stretching and possibly N-H bending vibrations. Further, the peak observed at 1105.01 cm^−1^ indicated the presence of strong C-O bond stretching due to the presence of secondary alcohol. The occurrence of aliphatic organo-halogen compounds can be suggested due to a peak in the range of 1010 to 1005.01 cm^−1^ indicating C-halide bond stretching vibrations. Similarly, a peak at 1398.41 cm^−1^ can be ascribed to C-H bond vibration of alkene compounds. A small shoulder peak at 1511.44 cm^−1^ arising due to vibrations of the N=O and C-C bonds suggests the presence of nitro- and aromatic compounds. The bending vibration peaks (at 869.81, 848.0, and 829.4 cm^−1^) for C-H bond in aromatic compounds can also be observed in the nettle leaf extract FT-IR spectra. The FT-IR spectra for CuO and ZnO NPs showed an increase in the number of absorption peaks possibly due to the functionalization of NPs with bioactive compounds of the nettle leaf extract. The spectrum of CuO NPs and ZnO NPs showed absorption band at 3310 cm^−1^ which corresponded to the presence of the amine group. The peak at 1103 cm^−1^ for CuO NPs was similar to the peak observed in nettle leaf extract and thus, indicating the presence of secondary alcohol. The peak at 1636 cm^−1^ arising due to C-N bond (amine group) vibration was also recorded in the FT-IR spectrum of CuO NPs. A band at 673 cm^−1^ (alkyl halide) was observed in the FT-IR spectra of both CuO and ZnO NPs.

The TEM study was performed to obtain the morphological details including the size dimensions and shapes of the zinc and copper oxide NPs. The biosynthesized NPs showed variable morphologies. Spherical and irregular shaped CuO and ZnO NPs were recorded (Figure 1C). The average size of the phyto-synthesized CuO and ZnO NPs ranged between 10 to 50 nm and 50 to 100 nm respectively.

The DLS analysis of the aqueous CuO and ZnO NPs suspensions revealed their mono-disperse nature with pdi of 0.142 and 0.195 for CuO and ZnO NPs respectively. The NP size distribution curve showed dimensions ranging from 5.13 to 21.48 nm for CuO NPs and 50.13 to 80.57 nm for ZnO NPs while the average hydrodynamic diameters were 11.42 ± 2.5.0 nm and 65.24 ± 3.1 nm respectively (Figure 1D). Both CuO (ζ = −27.0± 0.4 mV) and ZnO NPs (ζ = −23.0 ± 0.3 mV) aqueous suspensions had negative values.

### 3.2. DPPH Radical Scavenging Activity (RSA)

Nettle leaves possess remarkable biological activities. The FT-IR spectroscopy study of nettle leaf extract showed that these are the potent source of phenolic compounds, ascorbic acid, chlorophyll content, and carotenoids [20,23]. These compounds possess remarkable antioxidant activity and function as biotemplating and reducing agents for the generation of NPs. The phyto-synthesized CuO NPs exhibited significantly higher RSA as compared to the nettle leaf extract (Figure 2) while the ZnO NPs possessed antioxidant activity statistically at par with the nettle extract. The order of DPPH RSA was CuO NPs > BHT ≥ ZnONPs/nettle extract.

### 3.3. Antimicrobial Activity of Synthesized NPs

The phyto-synthesized copper and zinc oxide NPs exhibited substantial AMA as evidenced by the formation of a zone of inhibition on lawn cultured bacterial growth. A significantly higher AMA against all the pathogens was recorded for CuO NPs as compared to ZnO NPs and the nettle extract (Figure 3a). However, the AMA was recorded to be at par or numerically low as compared to the antibiotic (chloramphenicol- 30 mcg). The CuO NPs exhibited maximum activity against *Campylobacter* (Figure 3b) followed by against *Salmonella* and *S. aureus* with least AMA against *E. cloacae* MTCC 509.

The nettle-leaf extract derived ZnO NPs showed maximum AMA against *S. aureus* (ZOI-24 mm) followed by *Campylobacter* and *Salmonella* (ZOI-15 mm). The least AMA was observed against *E. cloacae* (ZOI-14 mm). The plant extract of stinging nettle, as well as the commercial ZnO NPs at working concentrations similar to the nettle leaf extract derived ZnO and CuO NPs showed no inhibitory effect on any of the test pathogens. However, ZnO NPs showed less AMA as compared to chloramphenicol (Figure 3b). Thus, the nettle leaf extract derived NPs exhibited appreciable AMA and potential as antimicrobial agents. Further, the CuO NPs showcased comparatively higher AMA than ZnO NPs. The improved AMA of the CuO NPs can be attributed to their small size and greater ROS producing potentials compared to ZnO NPs.

### 3.4. AMA of Nanocomposite Films

The nanocomposite films containing nanoparticles as the filler agents could enhance the storage span of the packaged perishables due to associated antimicrobial potentials. The films synthesized by the incorporation of phyto-synthesized NPs showed clear ZOI (mm) against the test pathogens. On the contrary, the control chitosan film without any NPs showed the least inhibition as compared to nanocomposite film against all the test microorganisms. The CuO NPs-chitosan film exhibited maximum inhibition against all the test microorganisms with the highest inhibition zone (15 mm) against *E. cloacae* MTCC 509 followed by against *Salmonella* (13 mm), *S. aureus* (12 mm), and *Campylobacter* (11 mm) (Figure 4). On the other hand, ZnO NPs-chitosan films inhibited *Salmonella* (11 mm) growth only. Thus, it seems that the AMA potential of CuO NPs remained effective even on embedding in the chitosan polymer as compared to the ZnO NPs-chitosan films.

### 3.5. Physical Properties of the Nanocomposite Films

#### 3.5.1. Film Thickness

The average film thickness of both the chitosan and chitosan nanocomposite films was 10 µm. As these films were prepared by solvent casting followed by peeling of the film, the films were not uniform along all the side and therefore, an average of ten thickness measurements were depicted. Further, the region of the film in contact with the mold had smooth surface while the other side was rough.

#### 3.5.2. Optical Properties of the Nanocomposite Films

##### Surface Color

The color of prepared films is influenced by the polymer and the NPs utilized for the fabrication of the films. The developed films possessed yellow color possibly due to the use of chitosan as the substrate (Figure 5a). The appearance of yellow color due to the addition of chitosan in chitosan-rice starch nanocomposite films has also been reported earlier [8]. The control chitosan film exhibited significantly the highest lightness (CIE L *) value (28.63) (Table 1). However, on the addition of CuO and ZnO NPs the L * value significantly decreased such that these films appeared darker than the control film. The lowest value of CIE a * was recorded with the CuO NPs-chitosan film. While a significantly higher chroma (3.22) caused the dark appearance of ZnO NPs-chitosan films and the lowest chroma value (1.51) gave light and dull appearance in the CuO NPs-chitosan film.

##### Film Transparency

The addition of CuO and ZnO NPs as nanofillers resulted in alteration in the transparency of the developed films. The calculated opacity value (T-value) was inversely proportional to transparency. The CuO NPs-chitosan film was the least transparent (T-value 1.99) (Table 1). The results are in agreement with the chroma value that was again lowest for CuO NPs-chitosan film indicating it to be less transparent. This may be due to the increased green appearance of these films on the addition of CuO NPs. However, a significantly lower value (T-opacity: 1.70) was observed for the ZnO NPs-chitosan film. Therefore, this film appeared more transparent.

##### Film Moisture

The film moisture content depends on the polymer used and the type of nanofiller added. Packaging films should possess lower moisture content if these are to be utilized for packaging food. The addition of nanoparticles significantly reduced the moisture level of the chitosan film (Table 2). Chitosan control film possessed the highest MC followed by chitosan-CuO NPs film. The lowest MC content was observed in the chitosan film containing ZnO NPs.

##### Water Holding Capacity

Degradable polymer-based packaging films tend to hold more water content i.e., possess high moisture content and are unsuitable for food packaging applications. Therefore, the development of nanocomposite films with improved water barrier properties and significantly reduced WHC or lesser MC are considered ideal for packaging as these characteristics help in preventing the growth of microorganisms. The addition of NPs significantly reduced the WHC of the chitosan film (Table 2). In this study, ZnO NPs addition in chitosan film significantly reduced the WHC (20.97%) followed by CuO NPs (26.09%). Therefore, the nanocomposite films are anticipated to exhibit improved water barrier property.

##### Film Solubility

Film Solubility (FS) also depends on the MC and WHC characteristics of the film. Chitosan control film exhibited significantly high FS% as compared to the nanocomposite chitosan films (Table 2). The films with higher FS tend to possess poor water barrier properties. Thus, an ideal packaging material must exhibit low water solubility. Chitosan-ZnO NPs film possessed the least solubility. 

### 3.6. Chemical Structure Characterization of the Nanocomposite Films

#### SEM and SEM-EDS of the Developed Films

Surface characteristics of films were influenced by the addition of NPs as seen in Figure 5a. The SEM study revealed that all the films developed had smooth surface morphology. However, few aggregations were observed in NPs incorporated films as compared to the chitosan polymer film. However, despite few aggregations, the Cu and Zn elements were recorded to be well dispersed within the matrix of films. The elemental composition was determined by EDS and showed the surface occurrence of Cu and Zn elements on the developed films (Figure 5b,i,ii). In the films, the amount of Zn element (0.82%) was more as compared to Cu (0.41%) element. The decrease in the element (% wt./atom) amounts for the fabricated films may be due to the embedding of the NPs in the polymer leading to lower X-ray signals.

### 3.7. Antioxidant Properties of Prepared Chitosan Nanocomposite Films

The DPPH radical scavenging activity of the developed films was evaluated. The prepared films showed substantially low DPPH radical scavenging activity compared to the activity of as-prepared ZnO and CuO NPs. This decrease may be attributed to the concealing or embedding of the NPs in the chitosan matrix. The order of the DPPH activity followed the trend CuO NPs-chitosan film > ZnONPs-chitosan film >chitosan film (Figure 6). The films were also evaluated for ABTS scavenging activity which also exhibited a similar trend.

### 3.8. Physiological, Biochemical, and Microbiological Characterization of Guava Fruits

#### 3.8.1. Weight Loss (%)

The physiological loss of weight of guava fruits increased significantly as the days of storage (DOS) passed (Figure 7A). At all the storage intervals, the highest weight loss was recorded in the fruits not packaged in any film (control group). The fruits packed in chitosan film alone exhibited less weight loss (5.7 and 10.9 percent after 14 and 21 DOS, respectively). However, the fruits stored in ZnO NPs-chitosan film showed the minimum weight loss (2.60% and 4.75% at 14 and 21 DOS). It was closely followed by CuO NPs-Chitosan film packed fruits.

#### 3.8.2. Fruit Firmness

The fruit firmness decreased from the start of the storage until the end of the 21 DOS (Figure 7B). The guava fruits wrapped with ZnO NPs-chitosan film maintained the highest fruit firmness (40.5 and 31.6 Newton at 14 and 21 DOS, respectively). It was followed by fruit firmness in CuO NPs-chitosan film (32.0 and 29.0 Newton at 14 and 21 DOS, respectively). The minimum fruit firmness was recorded in fruits packaged with chitosan film (alone) which did not differ from fruit firmness in the fruits stored without any film (control). A significant (*p* ≤ 0.01) correlation was observed between fruit firmness and weight loss (Table 3).

#### 3.8.3. Soluble Solids Content

Irrespective of the treatments, an initial increase in the SSC content was recorded at 7 DOS (Figure 7C) which later decreased by the end of the second week (at 14 DOS). After 3 weeks of storage (21 DOS), the highest SSC level (9.1%) was observed for unpackaged control fruits. It was closely followed by SSC level (8.5%) in fruits wrapped in CuO NPs-chitosan film which did not differ significantly from the SSC level in fruits stored in Chitosan films. The lowest SSC (6.8%) level was observed in fruits wrapped in ZnO NPs-chitosan film. 

#### 3.8.4. Titratable Acidity

The titratable acidity (TA) in the test guava fruits declined during the storage span (Figure 7D). At 14 and 21 DOS, the lowest TA levels (0.34 and 0.27%, respectively) were recorded in the unwrapped control fruits. The highest TA levels were recorded in fruits wrapped in ZnO NPs-chitosan film (0.42 and 0.37% at 14 and 21 DOS) which did not differ from the TA levels recorded in the fruits wrapped in chitosan films. It was closely followed by TA levels in the fruits stored in CuONPs-chitosan film.

#### 3.8.5. Sensory Quality

The sensory evaluation of the fruit (Figure 8A) at regular intervals revealed that the treatments significantly (*p* ≤ 0.05) affected the sensory quality (SQ) attributes viz., appearance, freshness, color, aroma, texture, and taste of the fruit. The SQ of the fruits increased up to 14 DOS and decreased by 21 DOS. The unwrapped control fruits had the highest SQ (8.30) after 7 DOS which was at par with SQ for the fruits packed in ZnO NPs-chitosan film. At 14th DOS, the highest SQ (8.5) was observed for the fruits stored in CuO NPs-chitosan film which did not significantly differ from the SQ values for the fruit stored in ZnO NPs-chitosan film and unwrapped fruit. At the end of three weeks of storage, the fruit packed in the chitosan-NPs films maintained the highest SQ (≈8.1) while the unwrapped fruits had the lowest SQ (6.98).

#### 3.8.6. Fruit Spoilage

The guava fruits stored under refrigerated conditions exhibited no spoilage until 7 DOS in any of the treatments (Figure 8B). However, by the end of the second week of storage, unwrapped control fruit exhibited 7.3% spoilage followed by a 2.1% spoilage in the fruit wrapped in chitosan. After 14 DOS, no spoilage was observed for the fruits packed in both the chitosan-NPs films. At the end of 21 DOS, the lowest spoilage (6.3%) was recorded in the fruits wrapped in ZnO NPs-chitosan film which did not differ significantly from the spoilage recorded in the fruit stored in CuO NPs-chitosan film. Unwrapped control fruit exhibited the highest spoilage (22.9%) percent followed by fruit packed in chitosan film (16.7% spoilage). The higher incidence of spoilage in the unwrapped fruit over chitosan-NPs film might be due to higher SSC and microbial load on the fruit (Figure 9). Hence, the NPs addition in chitosan polymer films significantly reduced spoilage evident from lower microbial viable cell counts in the fruit wash water. A strong correlation was observed among the physiological and sensory attributes of guava fruits for the spoilage (%), firmness and weight loss (%) characteristics (Figure 10A–C). Further, the highest spoilage in the unwrapped fruits might be due to higher weight loss as spoilage had a significant positive correlation (0.895, R^2^ = 0.802) with the weight loss in the fruit (Figure 10A). Besides, a significant negative correlation exists between the spoilage and fruit firmness (−0.626, R^2^ = 0.392) which suggested that the higher reduction loss in fruit firmness might be responsible for higher spoilage in the unwrapped fruits (Figure 10B). A significant correlation between fruit firmness and weight loss (−0.848, R^2^ = 0.719) was observed which suggested that the greater loss in firmness in unwrapped fruit might be due to higher weight loss in the fruit (Figure 10C).

Microbial spoilage enumerated as viable counts of TAB, yeast, mold, and Enterobacteriaceae was determined as log colony-forming units per mL (log cfu/mL). The viable count was significantly increased with the days of storage (Figure 9). After 21 DOS, unwrapped fruits showed the highest microbial count which can be due to increased exposure to the storage environment. The highest bacterial count (6.66 log cfu/mL) was observed in the wash-waters of the control fruits as determined on nutrient agar medium. While the lowest count of the TAB (6.45 log cfu/mL) was observed for ZnO NPs-chitosan film packaged fruits followed by CuO NPs-chitosan film packaged fruit (6.52 log cfu/mL) (Figure 9a). The TAB enhanced corresponding to time-span of storage. 

The counts of yeast cells were highest and lowest for control and chitosan-ZnO NPs film packaged fruits respectively (Figure 9b). Moreover, the highest mold counts (6.48 log cfu/mL) were recorded for control and chitosan film packed fruits. However, the lowest mold counts were recorded on the fruits stored in the ZnO NPs-chitosan film. The count of Enterobacteriaceae family members was highest on control fruits and lowest counts.

## 4. Discussion

The synthesized NPs were characterized by different spectroscopy and microscopy techniques. The UV-Vis spectra of the CuO NPs exhibited a similar SPR peak at 380 nm as reported for the formation of CuO NPs using the *Gloriosa superb* L. extract [31]. Likewise, UV absorption peaks at 380 nm for the *Passiflora caerulea* leaf extract derived ZnO NPs have also been reported [32]. Moreover, similar SEM observations regarding the synthesis of aggregates as well as individual CuO NPs using leaf extract of *Albizia lebbeck* have been reported [33] with the presence of Cu element (34.78% weight) on EDS analysis. In contrast to this, ZnO NPs derived from the leaf extract of *Passiflora caerulea* showed the presence of a higher percentage (75.36%) of the Zn element [32].

Stinging nettle leaves contain a significant amount and diversity of low molecular weight primary biomolecules such as chlorophyll, carboxylic acids, carbohydrates, sterols, polysaccharides, essential amino acids, fatty acids, vitamins, and biologically active compounds such as terpenoids, carotenoids, tannins, and isolectins [21]. The FT-IR spectroscopy was performed to analyze diverse functional groups (minerals, alcohols, ethers, esters, aliphatic/alkenes, compounds with the aromatic ring as phenols and nitro compounds) present in the nettle extract and the phyto-synthesized CuO and ZnO NPs [34]. The occurrence of a broad peak (3324.68 cm^−1^) for the hydroxyl group indicated the presence of alcohol and phenol [35]. The C-H stretching vibration (2970 cm^−1^) corresponded to the presence of an aliphatic (alkane) group [36]. A similar peak at 2918.20 cm^−1^ has been reported in the FT-IR spectrum of nettle leaf extract [37]. The 1636.30 cm^−1^ peak suggested the presence of carbonyl and amine functional groups [38]. The occurrence of alkyl halide [C-halide] (C-F/C-Br) bond bending vibrations can be identified due to the presence of multiple peaks in the range of 688.5 to 670.8 cm^−1^ [35]. The presence of similar peaks (though broader and substantially subdued) and thus, the same bioactive compounds in phyto-synthesized NPs were also identified indicating the functionalization of the NPs with organic compounds present in nettle leaf extract [35]. Similarly, peaks at 1032, 1125, and 1382 cm^−1^ representing the presence of alcohol were observed in ZnO NPs synthesized from aqueous extract of Oak Fruit Hull (Jaft) [38]. The peak at 673 cm^−1^ (alkyl halide) suggests the metal-O bond bending vibrations [34]. Thus, similar peaks with the nettle extract had depicted that synthesized metal oxide NPs have been functionalized with the bioactive functional groups of nettle extract from which they are synthesized and act as a stabilizing/capping agent.

The TEM results obtained for the CuO and ZnO NPs which revealed spherical to semi-spherical morphologies of the NPs were similar to another report which documented the synthesis of spherical shaped 5–10 nm sized CuO NPs using leaf extract of *G. superb* [31]. Likewise, *Euphorbia jatropa* latex was utilized as a reducing agent to generate ZnO NPs of a size range of 50–200 nm [39]. However, the synthesis of ZnO NPs using *Moringa oleifera* had a size of less than 50 nm [40]. The variation in the size dimensions among phyto-synthesized NPs may be attributed to variability in the biomolecules occurring in the plant extracts. The phenomena of occurrence of higher hydrodynamic diameters for the synthesized nanoparticles have also been reported by Kalia et al. [18] for citrus callus and fruit juice derived silver nanoparticles. Likewise, similar negative zeta potential values have been reported for the citrus callus and juice derived AgNPs [18].

Nettle leaf extract is rich in several phytochemicals and compounds having high antioxidant potential [41]. A higher RSA has been reported for silver NPs generated from leaf extract of *Chenopodium murale* than the aqueous leaf extract [42]. Furthermore, *Solanum nigrum* leaf extract derived ZnO NPs also exhibited higher RSA than the leaf extract [43]. However, among the two biosynthesized NPs, CuO NPs exhibited higher RSA as compared to ZnO NPs. The results regarding the AMA of the nettle leaf extract derived CuO and ZnO NPs share the results reported for the *Ocimum basilicum* leaf extract synthesized CuO NPs which exhibited appreciable AMA as indicated through the zone of inhibition formed against *S. aureus* and *Escherichia coli* (7.2 mm and 9.8 mm respectively) but the leaf extract control showed no inhibition zone [44]. Another study evaluating the antibacterial activity of Ag NPs synthesized from *Aloe fleurentiniorum* extract reported the formation of distinct zone of inhibition against the test organisms. However, the plant extract which served as a control showed no zone of inhibition [45].

The chitosan-CuO/ZnO nanocomposite films were also characterized for various film parameters. The chitosan-rice starch and chitosan-rice starch nanocomposite films exhibited similar film thickness (average 10 mm). These films were fabricated for improving the storage shelf-life of peach fruit [8]. Contrary to this, another published research work on the use of Ag NPs as nano-filler agents in chitosan film reported an increase in the film thickness compared to the control film [46]. The prepared films exhibited the appearance of yellow color. Yellow-appearing films were also obtained by the addition of chitosan in chitosan-rice starch nanocomposite films [8]. Likewise, another report recorded a decrease in the value of L * on the addition of Ag NPs in the chitosan film [46]. The incorporation of the NPs improved the transparency attribute of the prepared films. Likewise, the decrease in the transparency of the chitosan films by incorporation of Ag NPs has been reported [46]. Among the other physical characteristics of the prepared films, the addition of nanofillers reduced the moisture content of the films. A similar reduction in the film moisture content has been reported on incorporation of Ag NPs and MMT in the PVA film [47], and AgNPs and cellulose nanocrystal incorporation in chitosan polymer film [48]. Moreover, the film solubility properties were altered particularly reduction in film solubility was observed on incorporation of the ZnO NPs. This could be due to intercalation of the ZnO NPs within the polymer matrix leading to formation of strong hydrogen bonds between the amide and glycosidic functional groups of chitosan with the ZnO that did not allow for easy solubilization of the polymer matrix on water immersion. A similar trend has been reported in a recent study in which the addition of MMT and Ag NPs within the PVA film helped to reduce the FS percent [47].

The SEM and SEM-EDS studies of the prepared films exhibited occurrence of smooth surface with few aggregations. A similar study of SEM analysis of chitosan-gelatin polymeric film showed little aggregations on the surface of the film due to the addition of ZnO NPs [49]. Another research work on SiO_2_ NPs incorporated starch films depicted the smooth surface of the developed films [50]. Moreover, the chitosan-rice starch nanocomposite film showed development of films having a smooth surface [8]. The antioxidant properties of the chitosan nanocomposite films revealed comparatively better activity for the CuO NPs-chitosan films. Though the antioxidant activities were relatively low compared to the NPs formulations, still these results can be corroborated with the earlier published reports on chitosan-based composite films containing cinnamon oil [51], cinnamaldehyde [52], and nano-TiO_2_-Clove oil [53] exhibiting similar antioxidant properties of chitosan and chitosan nanocomposite films.

The effect of packaging of guava fruits in chitosan nanocomposite films was assessed based on various fruit quality attributes. Fruit weight loss (%) is a common fruit quality trait. Primarily evapotranspiration and respiration are the major causes of the loss of weight of the fruits during storage [54]. However, the weight loss in guava which exhibits climacteric behavior, postharvest ripening process continues leading to ethylene hormone-induced degradation of the hemicellulose content in the fruits and fruit softening. Application of chitosan alone as fruit coating retards ripening in guava due to suppression of the respiration rate besides inducing antioxidant activities through significant enhancement in the peroxidase activity [55]. The incorporation of the ZnO and CuO NPs as filler agents in chitosan polymer must have further improved the gas barrier properties of the polymer thereby decreasing the respiration rate, and ethylene-induced ripening effects due to elicitation of the fruit’s native antioxidant activities. The decreased weight loss can also be indirectly attributed to maintenance of the acid content (titratable acidity) in the nanocomposite film packaged fruits (Figure 7D, Table 3). It can also be identified that there exists a direct relationship among the moisture content and the film solubility. The films with high MC exhibit high WHC and FS and low water barrier properties. The ZnO/CuO NP-chitosan films improved the water and gas barrier properties therefore, these led to decrease in the MC, WHC and FS properties of the developed chitosan nanocomposite films.

The fruit firmness is influenced by both the evapotranspiration and respiration rates during storage. Therefore, the loss of solutes and water from the fruit surface must be considered to be responsible for loss of firmness. However, as indicated above in the description for the fruit weight loss (%) parameter, the predominant cause for the loss of firmness in guava is ethylene-induced degradation of the hemicellulose content [56]. The significant positive correlation of titratable acidity with fruit firmness further confirms the elaborate biochemical transformations which were delayed in nano-composite film packaged fruits and helped in maintenance of high titratable acidity compared to unpackaged or chitosan alone packaged fruits. Collectively, transpiration from the cuticle results in weight loss, and degradation of the cell wall by enzymes viz., pectin methylesterase, and cellulose finally resulted in decreased fruit firmness during storage. Similar results have been reported for the Chinese jujube fruits wrapped with polyethylene composite containing silver, titanium dioxide, kaolin NPs which showed a delay in fruit firmness in comparison to unpackaged fruit and the fruits wrapped in the film without NPs [57].

Soluble solids content (SSC) is an important quality parameter that exhibits a direct correlation with the texture and composition of the fruit [58]. The increase in SSC might be due to metabolic transformations of the soluble compounds such as sugars [59]. During the ripening process, starch is utilized as a carbon source for the generation of sucrose and other aroma-specific volatile compounds [60]. As guava is a climacteric fruit, higher SSC may indicate rapid ripening of the fruits during storage. Thus, ZnO NPs-chitosan film and chitosan film might have reduced the rate of ripening process which may have resulted in the lower SSC levels at 21 DOS over control fruits. A rapid loss in TA was recorded in unwrapped control fruits which might be due to enhanced malic enzyme activity and pyruvate decarboxylation during the storage period [61]. Further, the decrease in TA over the storage time might be ascribed to the consumption of organic acids in metabolic activities such as respiration. Besides, the decrease in TA at the later stages of fruit development may also be due to the transformation of organic acids into sugars [62] as reflected by higher SSC levels in unwrapped control (Figure 7C). The fruit spoilage (%) was decreased with the fruit wash water studies which illustrated an overall low microbial viable count for nanocomposite film packaged fruits. The anti-mycotic and anti-bacterial potential of the CuO and ZnO NPs can be attributed to formation of reactive oxygen species (ROS) and the metallic ion obtained on dissolution of the NPs in the cell milieu after internalization of the NPs in the cell [63]. However, as a function of the length of the storage duration, an increase in the bacterial and fungal counts was reported on storage [64]. The fresh-cut papaya slices wrapped in chitosan-ZnO nanocomposite coating exhibited lower microbial counts as compared to the control fruits [65]. Likewise, metal or metal oxide nanoparticles have also been utilized to check the growth of microbes causing spoilage of a variety of fruit-based beverages particularly the fruit juices [66].

## 5. Conclusions

The metal oxide NPs synthesized from the nettle leaf extract exhibited characteristic morphologies. The AMA of these NPs against test pathogenic cultures was higher compared to the nettle leaf extract. The CuO NPs exhibited better efficacy considering their highest AMA which may be ascribed to size-dependent phenomena as CuO NPs had smaller size dimensions (10–50 nm) compared to the ZnO NPs (50–100 nm). On DLS analysis, the nanoparticles showed slightly larger hydrodynamic size distribution which spanned over 5.13 to 21.48 nm and 50.13 to 80.57 nm for CuO NPs and ZnO NPs respectively. The as-prepared nanosols possessed functional groups present in the nettle leaf extract which acted as a surface priming agent for the NPs in FT-IR analysis.

Further, the incorporation of these NPs in chitosan films improved the physical characteristics of the films. The addition of CuO NPs in chitosan film has improved the antimicrobial potential more effectively than the ZnO NPs. Additionally, fruits packaged with nanocomposite films were reported with the least microbial load and spoilage in comparison to unpackaged fruits NPs. The developed films were transparent, had low antioxidant but substantial AMA with CuO NP-composite film having significantly higher AMA. The microbial count was comparably equal on the surfaces of the fruits packaged with chitosan films containing CuO and ZnO NPs. However, it was less as compared to control fruits as well as to fruits packaged with control chitosan films without NPs. Therefore, these nanocomposite films having biologically synthesized NPs as nano-fillers can be used to improve the shelf life and quality of various fruits more effectively than the polymeric films without the incorporation of any nanomaterials.

## Figures and Tables

**Figure 1 biomolecules-11-00224-f001:**
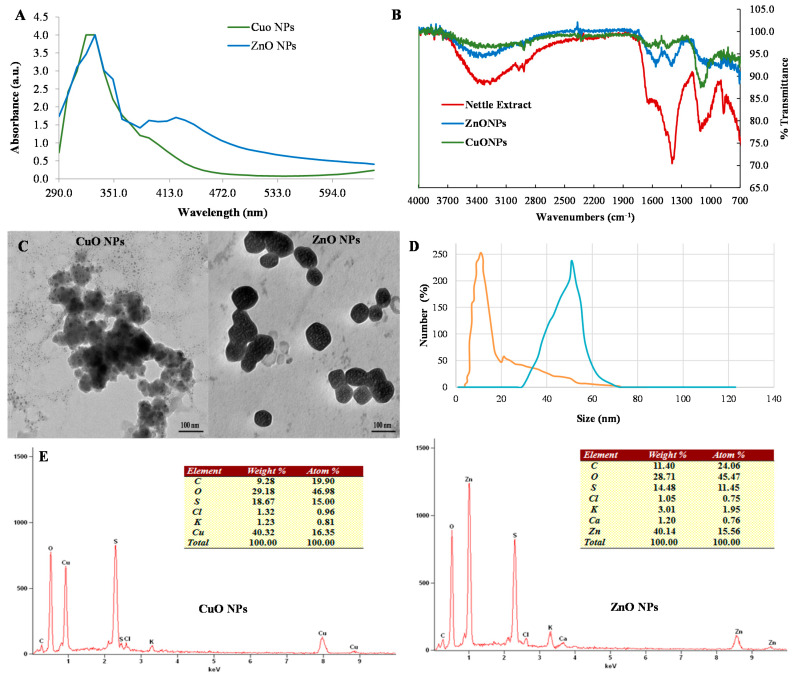
Electron microscopy and spectroscopy characterization of the phyto-synthesized CuO and ZnO nanoparticles, (**A**) UV-Vis spectra, (**B**) FT-IR spectroscopy analysis, (**C**) Transmission electron microscopy analysis, (**D**) Dynamic light scattering spectroscopy, and (**E**) SEM-EDS (Energy dispersive spectroscopy) spectra.

**Figure 2 biomolecules-11-00224-f002:**
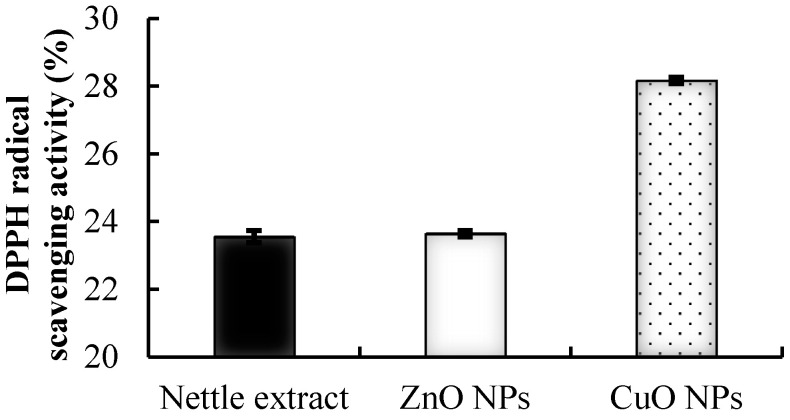
Antioxidant activity of the synthesized nanoparticles analyzed as percent DPPH radical scavenging activity.

**Figure 3 biomolecules-11-00224-f003:**
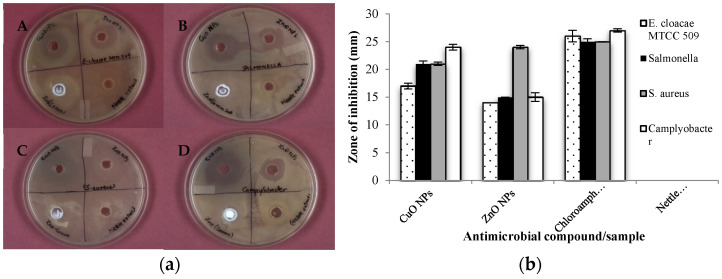
Antimicrobial activity of the synthesized NPs against (**a**) Optical images depicting AMA against (A) *E. cloacae* MTCC 509, (B) *Salmonella*, (C) *Staphylococcus aureus*, and (D) *Campylobacter*; (**b**) Graphical representation depicting comparative AMA (ZOI in mm) of synthesized NPs, antibiotic, and nettle extract.

**Figure 4 biomolecules-11-00224-f004:**
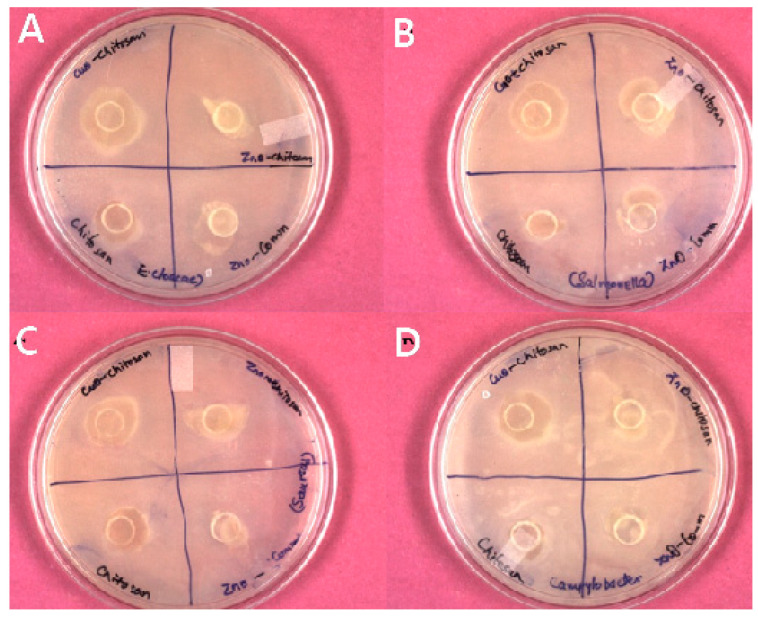
Antimicrobial activity of the chitosan nanocomposite films against (**A**) *E. cloacae* MTCC 509, (**B**) *Salmonella*, (**C**) *Staphylococcus aureus*, and (**D**) *Campylobacter.*

**Figure 5 biomolecules-11-00224-f005:**
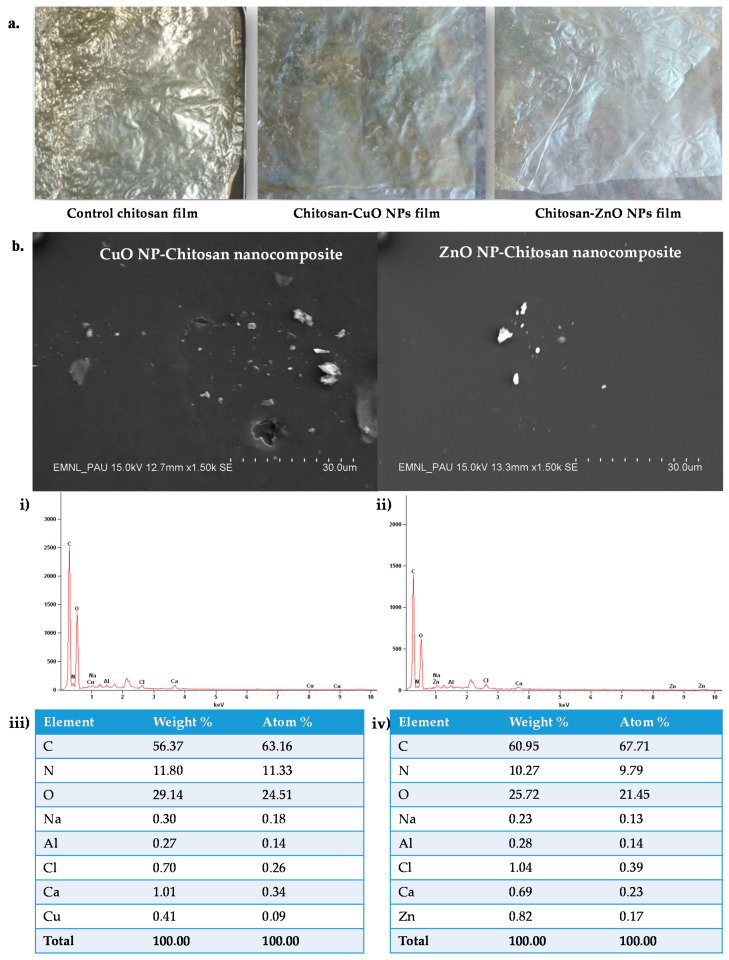
Optical images and Scanning electron micrographs of the synthesized chitosan nanocomposite films. (**a**) Optical images of chitosan control and nanocomposite films; (**b**) SEM image depicting the film surface and SEM-EDS spectra of the CuO and ZnO-chitosan nanocomposite films; (**i**) EDS spectra of Chitosan-CuO NPs films, (**iv**) (**ii**) EDS spectra of Chitosan-ZnO NPs nanocomposite film, (**iii**) Table depicting EDS quantitative results for Chitosan-CuO NPs films, (**iv**) Table depicting EDS quantitative results for Chitosan-ZnO NPs films.

**Figure 6 biomolecules-11-00224-f006:**
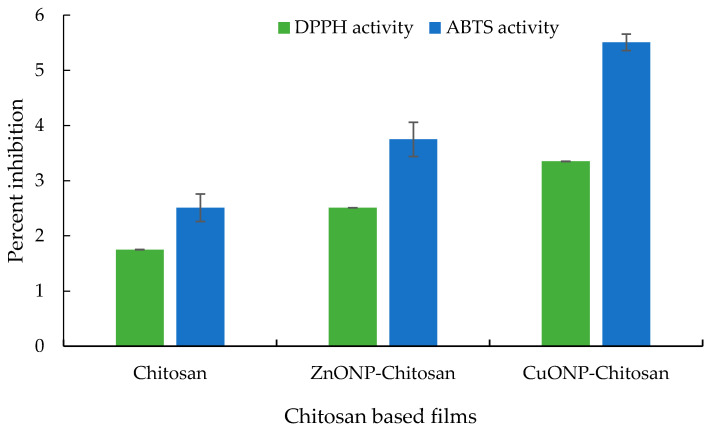
Antioxidant activity of the chitosan nanocomposite films.

**Figure 7 biomolecules-11-00224-f007:**
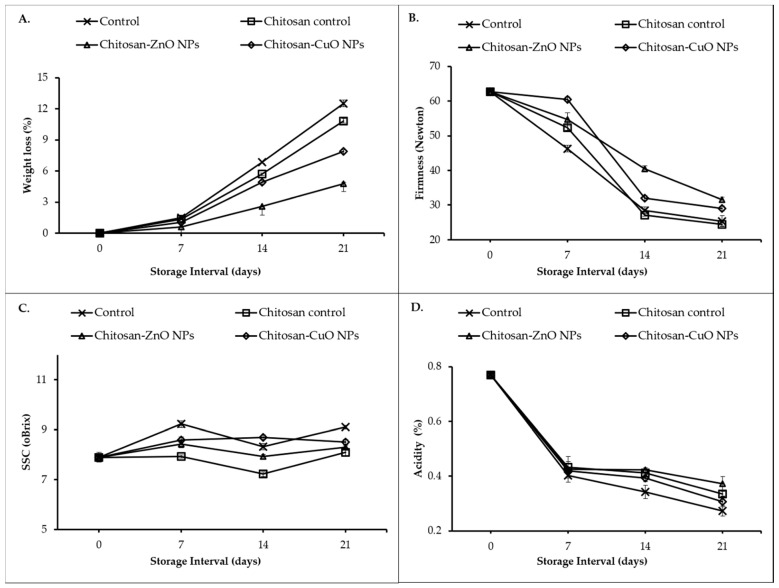
Effect of nanocomposite film packaging on physiological and biochemical parameters of guava fruits. (**A**) Weight loss percentage; (**B**) Fruit firmness; (**C**) Soluble solid content; (**D**) Acidity.

**Figure 8 biomolecules-11-00224-f008:**
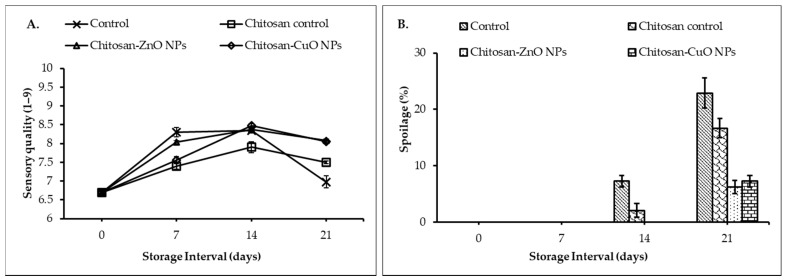
Sensory quality evaluation and fruit spoilage (%) determination at different days of storage. (**A**) Sensory quality of packaged fruits evaluated through a nine-point hedonic scale; (**B**) Percent spoilage of packaged fruits.

**Figure 9 biomolecules-11-00224-f009:**
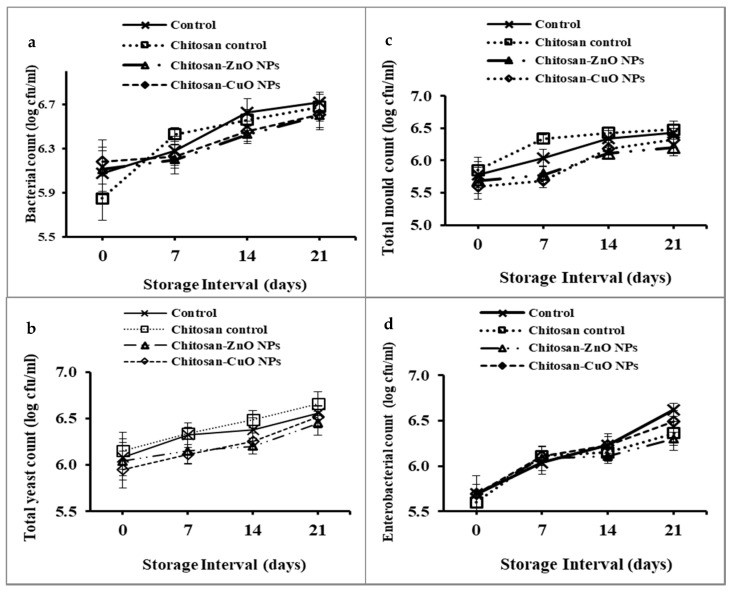
Microbial viable counts (log cfu/mL) of the wash water of guava fruits packaged in chitosan/chitosan nanocomposite films at different days of storage. (**a**) Total bacterial count, (**b**) yeast count, (**c**) mold count and (**d**) enterobacterial count.

**Figure 10 biomolecules-11-00224-f010:**
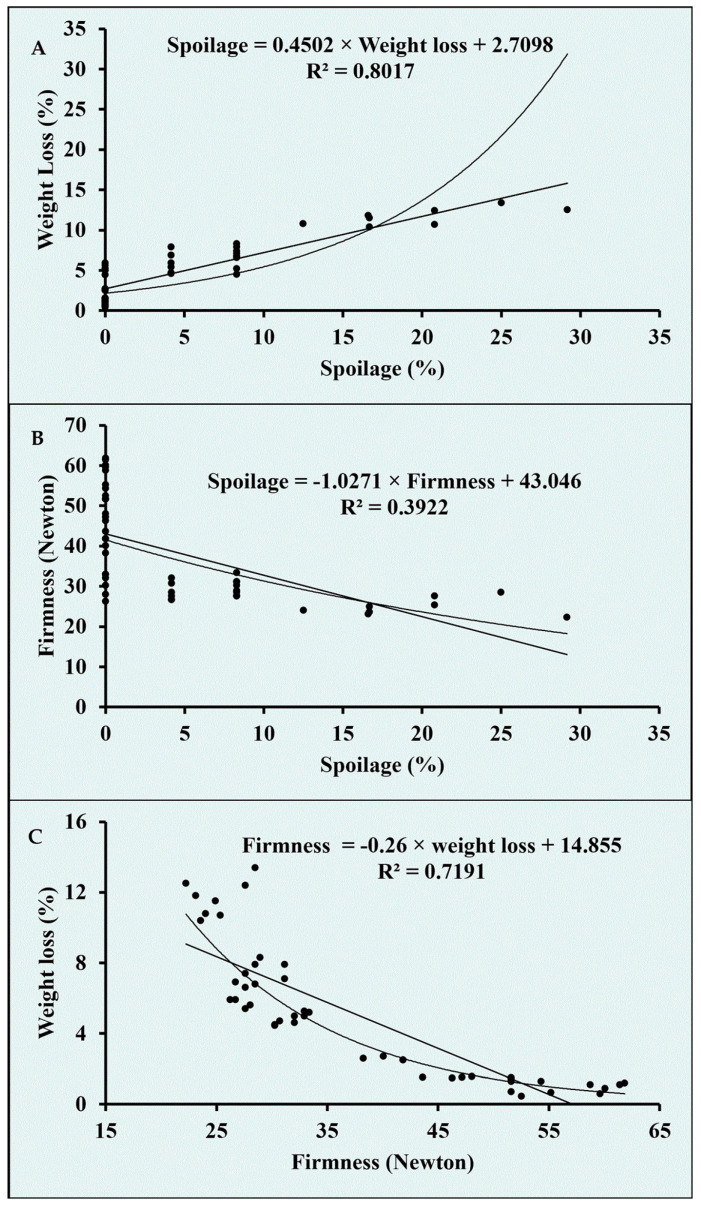
Correlation graphs between physiological and sensory quality parameters of Guava. (6.2 log cfu/mL) were observed on the fruit surface packaged with CuO NPs-chitosan film. (**A**) Correlation between spoilage (%) and weight loss (%), (**B**) Correlation between spoilage (%) and firmness, (**C**) Correlation between firmness and weight loss (%).

**Table 1 biomolecules-11-00224-t001:** Lightness (L), chroma, hue and opacity (T) value of nanocomposite polymer film and control chitosan film.

Types of Films	L *	a *	b *	C *	h°	T (Opacity)
Control chitosan	27.58 ± 0.82 ^a^	−0.45 ± 0.07 ^b^	2.36 ± 0.18 ^b^	2.35 ± 0.07 ^b^	100.89 ± 0.33 ^c^	1.95 ± 0.03 ^b^
Chitosan-CuO NPs	25.55 ± 1.28 ^b^	−0.88 ± 0.09 ^a^	1.23 ± 0.04 ^c^	1.54 ± 0.09 ^c^	125.47 ± 0.36 ^a^	2.01 ± 0.06 ^a^
Chitosan-ZnO NPs	27.48 ± 0.92 ^a^	−0.76 ± 0.06 ^b^	3.33 ± 0.31 ^a^	3.37 ± 0.24 ^a^	103.68 ± 0.25 ^b^	1.70 ± 0.13 ^c^

All the values are depicted as mean (*n* = 3) ± S.E., values in the column followed by the same superscripted letter are not significantly different, L *: lightness, a * and b *: chromaticity coordinates, C *: chroma, h°: hue angle.

**Table 2 biomolecules-11-00224-t002:** Moisture content (MC), water holding capacity (WHC) and film solubility (FS) of chitosan/chitosan nanocomposite films.

Film Type	MC (%)	WHC (%)	FS (%)
Chitosan control	23.87 ± 1.86 ^a^	63.41 ± 2.28 ^a^	30.96± 2.11 ^a^
Chitosan-ZnO NPs	5.16 ± 1.62 ^b^	20.97 ± 1.69 ^b^	10.59 ± 1.33 ^b^
Chitosan-CuO NPs	6.15 ± 0.38 ^b^	26.09 ± 0.66 ^b^	13.58 ± 0.95 ^b^

All the values are depicted as mean (*n* = 3) ± S.E., values in the column followed by the same superscripted letter are not significantly different.

**Table 3 biomolecules-11-00224-t003:** Person’s correlation between among quality attributes in guava.

	Firmness	SSC	TA	SQ	Weight Loss	Spoilage
Firmness	1					
SSC	0.197	1				
TA	0.597 **	−0.193	1			
SQ	−0.035	−0.104	0.262	1		
Weight loss	−0.848 **	0.084	−0.721 **	−0.378 **	1	
Spoilage	−0.626 **	0.146	−0.700 **	−0.539 **	0.895 **	1

** Correlation is significant at the 0.01 level (2-tailed).

## Data Availability

Data is contained within the article.

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
