# Peer review of "Nettle-Leaf Extract Derived ZnO/CuO Nanoparticle-Biopolymer-Based Antioxidant and Antimicrobial Nanocomposite Packaging Films and Their Impact on Extending the Post-Harvest Shelf Life of Guava Fruit"

_biomolecules, 2021, doi:10.3390/biom11020224_

Round 1

Reviewer 1 Report

A brief summary

The CuO or ZnO nanoparticles, which was prepared from the green synthesis by using stinging nettle leaf extracts, possessed certain antioxidant and antimicrobial activity. The chitosan film with CuO or ZnO nanoparticles was then applied to extend the shelf-life of the packaged guava fruits.

Broad comments

Green packaging films with metal oxide nanoparticles biosynthesized by leaf extract, including functionalized by stinging nettle leaf extracts, have been studied and been published in the recent few years. Therefore, this research is not novel.

So far five nanomaterials have been approved by the European Food Safety Authority (EFSA) for food contact materials as they showed no safety concern regarding human health: titanium nitride, selenium nanoparticle, silica, montmorillonite clay, and zinc oxide. While particles reach nanoscale, specific properties of nanoparticles (NPs) may cause more hazard potential to health and environment. Among these common metal oxides NPs, the toxicity of CuO NP is highest. Some studies had reported that after pinocytosing CuO NPs, cells showed specific lethality because of reactive oxygen species (ROS) induced by NPs and some still unknown mechanisms from NPs surface. Furthermore, oxidative stress inducing from ROS and Cu+2 released from CuO NPs would cause DNA oxidative damage. Therefore, the results in this research showed that the RSA and AMA of CuO are better than those of ZnO, which can be expected. The authors planned to use CuO NP with high bactericidal ability as guava packaging, its safety must be carefully considered. The authors only mentioned in the "Introduction, L.81-83" that "the fabrication of natural polymer-based nanocomposite films low cost and ecologically safe nanoparticles are required." The authors did not mention the possible health risks of metal oxide.

There are many research methods in the manuscript that did not clearly state the experimental conditions or statistical methods of experimental data. Although the authors have conducted a lot of experiments, the data have not been intensive discussion or the inferences are irrational, more reasonable explanations and inferences should be provided.

Specific comments

P.123

What is the molecular weight range of chitosan? The molecular weight will affect the film forming ability and the physical properties of the film.

P.140-151

The instructions for the synthesis of CuO and ZnO NP are not detailed enough, including whether the authors regulated the reaction pH, temperature, reaction end point (influencing the formation of CuO and ZnO NP concentration), etc.

How to judge that CuO NP was obtained by visualizing the color change of the solution which turned green if stinging nettle leaf extract appears in color (it should be a little green)?

P.194

What is the ratio of chitosan and synthesized NP solution for fabrication of chitosan nanocomposite films?

P.229

It should be: Chroma (C) = (a*2 + b*2) ½

P.287

What is the size and color of guava for storage experiment? What is the maturity of guava?

P.299

There is no statement for the measure method in the firmness of guava sample after peeling. What were the operating conditions of penetrometer during testing?

P.317

In the sensory evaluation, the sensory quality scoring method is not stated, and the hedonic test of only 5 panels is not enough to provide accurate evaluation results. The evaluation included texture and taste, the samples were needed to be chewed in the mouth. As it relates to whether the guava was spoilage during storage, the food safety issues of the panels should be considered.

P.326

How to define the spoilage of guava?

P.335

This study used the phytochemicals act as a reducing agent for metal/metal oxide salts leading to the generation of NPs in nettle leaf. The authors used many indirect analysis methods to evaluate the production of CuO and ZnO. In fact, more direct or accurate Instruments (such as mass spectrometry, X-ray...) can be used to qualitatively or quantitatively analyze the reduced NP from zinc acetate and copper sulfate was CuO or ZnO, and was not other nanocopper, nanozinc or other compounds.

P.431

What is the final concentration of CuO and ZnO extracted with nettle leaf? How to avoid the agglomeration of these NPs? What concentration of CuO and ZnO were used for RSA and AMA experiments?

When preparing chitosan composite film, CuO and ZnO solution contained nettle leaf extract, how to evaluate the anti-oxidation and antibacterial ability of CuO, ZnO and film without the interference from nettle leaf extract?

P.482

It should be explained why the film was tested for its antibacterial activity against these four pathogens: Campylobacter, E. cloacae, Salmonella, and S. aureus? These bacteria are not spoilage bacteria of guava. What is the significance of the film to delay the spoilage of guava?

P.488

For the casting method, it is unlikely that these films appeared to be uniform along all the side. Due to the pulling effect of the surface tension of the solution and the adsorption force between the solution and the mold during the drying process, there are usually wrinkles around the film. And the side of the film in contact with the mold is smoother, and the other side is rougher.

P.520/P.597-599

It should be confirmed that Fig.5a is SEM or EDS?

P.604

Because the content of CuO and ZnO should be very low (otherwise there will be food safety concerns), the FTIR analysis of the nanocomposite films seems meaningless

P.638

Antioxidant properties of prepared chitosan nanocomposite films was not shown in Fig.4b.

P.679

The author concludes that greater loss in firmness in unwrapped fruit might be due to higher weight loss in the fruit. The firmness of fruit is mainly related to pectin and the structure of fruit matrix, the decrease in water content might lead to higher firmness. Because guava is a climacteric fruit, the authors should discuss the influence of these films on the ripening of guava in order to reasonably explain the physical properties of guava in Figure 6, Figure 7, and Table 3.

Author Response

Reviewer 1:

Comments and Suggestions for Authors

A brief summary

The CuO or ZnO nanoparticles, which was prepared from the green synthesis by using stinging nettle leaf extracts, possessed certain antioxidant and antimicrobial activity. The chitosan film with CuO or ZnO nanoparticles was then applied to extend the shelf-life of the packaged guava fruits.

 Broad comments

Green packaging films with metal oxide nanoparticles biosynthesized by leaf extract, including functionalized by stinging nettle leaf extracts, have been studied and been published in the recent few years. Therefore, this research is not novel.

Reply: We understand that the green chemical synthesis of metal oxide NPs and its use for polymer film development is not a novel process. However, we could not obtain any published reports on CuO NPs synthesis through stinging nettle.

So far five nanomaterials have been approved by the European Food Safety Authority (EFSA) for food contact materials as they showed no safety concern regarding human health: titanium nitride, selenium nanoparticle, silica, montmorillonite clay, and zinc oxide. While particles reach nanoscale, specific properties of nanoparticles (NPs) may cause more hazard potential to health and environment. Among these common metal oxides NPs, the toxicity of CuO NP is highest. Some studies had reported that after pinocytosing CuO NPs, cells showed specific lethality because of reactive oxygen species (ROS) induced by NPs and some still unknown mechanisms from NPs surface. Furthermore, oxidative stress inducing from ROS and Cu2+ released from CuO NPs would cause DNA oxidative damage. Therefore, the results in this research showed that the RSA and AMA of CuO are better than those of ZnO, which can be expected. The authors planned to use CuO NP with high bactericidal ability as guava packaging, its safety must be carefully considered. The authors only mentioned in the "Introduction, L.81-83" that "the fabrication of natural polymer-based nanocomposite films low cost and ecologically safe nanoparticles are required." The authors did not mention the possible health risks of metal oxide.

There are many research methods in the manuscript that did not clearly state the experimental conditions or statistical methods of experimental data. Although the authors have conducted a lot of experiments, the data have not been intensive discussion or the inferences are irrational, more reasonable explanations and inferences should be provided.

Reply: We thanks the reviewer for critically evaluating the manuscript. The discussion section has been altered and pooled.

Specific comments

Comment 1: P.123 What is the molecular weight range of chitosan? The molecular weight will affect the film forming ability and the physical properties of the film.

Answer: The range of molecular weight of the chitosan was of medium molecular weight (190,000 to 300,000 g mol-1).

 Comment 2: P.140-151 The instructions for the synthesis of CuO and ZnO NP are not detailed enough, including whether the authors regulated the reaction pH, temperature, reaction end point (influencing the formation of CuO and ZnO NP concentration), etc.

How to judge that CuO NP was obtained by visualizing the color change of the solution which turned green if stinging nettle leaf extract appears in color (it should be a little green)?

Answer: The synthesis of CuO nanoparticles was performed at room temperature and at neutral pH conditions. However, for the synthesis of ZnO NPs, the final pH was set at 9.0 using 1 N NaOH solution after drop-wise addition of the nettle leaf extract (pH=8.7).

The colour of stinging nettle extract was light brown and not green as the extract was obtained by boiling the leaves in deionized water. The image of the screw cap vial containing the nettle extract has been included here for clarity.  

Comment 3: P.194 What is the ratio of chitosan and synthesized NP solution for the fabrication of chitosan nanocomposite films?

Answer: The CuO and ZnO NPs were incorporated at the rate of 10 mg L-1 in the chitosan polymer to prepare the respective nanocomposite films.   

Comment 4: P.229 It should be: Chroma (C) = (a*2 + b*2) ½

 Answer: The correction has been incorporated.

Comment 5: P.287 What is the size and color of guava for storage experiment? What is the maturity of guava?

Answer: The average size of the guava fruits was 110-120 g per fruit. The fruits were harvested at firm green stage. At this stage, the fruit attains the desired size and the colour changes from dark to lighter green with a smoother skin.

Comment 6: P.299 There is no statement for the measure method in the firmness of guava sample after peeling. What were the operating conditions of penetrometer during testing?

Answer: The fruit firmness was determined with penetrometer (model no FT – 327, USA) fitted on manual test stand for uniform application of force. The firmness was measured after the removal of a thin slice of peel from opposite equatorial points on the fruit surface. The force applied to plunge the 8 mm stainless steel probe into the peeled guava flesh on the manual test stand was recorded and expressed in Newtons.

The description has also been added in the revised manuscript.

Comment 7: P.317 In the sensory evaluation, the sensory quality scoring method is not stated, and the hedonic test of only 5 panels is not enough to provide accurate evaluation results. The evaluation included texture and taste, the samples were needed to be chewed in the mouth. As it relates to whether the guava was spoilage during storage, the food safety issues of the panels should be considered.

Answer: The sensory evaluation of the fruit was judged on appearance, color, freshness, texture, aroma, and taste as per the hedonic scale (1–9). Where 1 = extremely undesirable, 2 = very much undesirable, 3 = moderately desirable, 4 = slightly undesirable, 5 = neither desirable nor undesirable, 6 = slightly desirable, 7 = moderately desirable, 8 = very much desirable, 9 = extremely desirable.

The panel of five judges (n = 5) was constituted on the basis of availability and experience. The judges were selected from the staff of Department of Fruit Science, Punjab Agricultural University having experience of fruit profiling. The group consisted of two females and three males aging 35-50 years. The fruit was divided in to eight slices and two slices were used for each evaluation but, the judges were free to ask for extra sample. On the fruits which were of eating quality on the basis of appearance, color and freshness were used for sensory evaluation.

Comment 8: P.326 How to define the spoilage of guava?

Answer: The spoilage of guava fruits can be identified as soft rot, occurrence of lesions on the fruit skin, fungal mycelial growth and growth of opportunistic human pathogenic microbes on the surface of fruits. The predominant postharvest spoilage pathogens of guava fruits include a variety of fungal pathogens such as Colletotrichum gloeosporioides, Rhizopus stolonifera, Fusarium oxysporum, Mucor sp., Aspergillus niger, A. fumigatus and A.  parasiticus. The most common postharvest soft rot of fruits is caused by A. niger.  

Comment 9: P.335 This study used the phytochemicals act as a reducing agent for metal/metal oxide salts leading to the generation of NPs in nettle leaf. The authors used many indirect analysis methods to evaluate the production of CuO and ZnO. In fact, more direct or accurate Instruments (such as mass spectrometry, X-ray...) can be used to qualitatively or quantitatively analyze the reduced NP from zinc acetate and copper sulfate was CuO or ZnO, and was not other nanocopper, nanozinc or other compounds.

Answer: Due to COVID-19 lockdown scenario, it was not possible to perform the XRD, and ICAP/ICP-MS analysis of the CuO/ ZnO NPs. However, as the nanoparticles were synthesized in ambient conditions and in aqueous medium, the formation of oxides of the Zn and Cu elements is the most likely proposition.  

Comment 10: P.431 What is the final concentration of CuO and ZnO extracted with nettle leaf? How to avoid the agglomeration of these NPs? What concentration of CuO and ZnO were used for RSA and AMA experiments?

Answer: The Cu and Zn concentration was 249.5 and 183.48 mg 100 mL-1 for the CuO and ZnO NPs respectively.

Generally, surfactant compounds (synthetic organic compounds such as poly vinyl pyrrolidone (PVP), polysorbate 20/ 40/ 60/ 80 and natural organic compounds such as saponins, polysaccharides) are added to improve stability of the prepared NPs.  

Both CuO and ZnO NPs were evaluated for RSA and AMA activities by using 10 mg L-1 as the working concentration.

Comment 11: When preparing chitosan composite film, CuO and ZnO solution contained nettle leaf extract, how to evaluate the anti-oxidation and antibacterial ability of CuO, ZnO and film without the interference from nettle leaf extract?

Answer: As the biomolecules present in the nettle leaf extract were used both as biotemplating and functionalizing molecules, it will not be possible to completely segregate the individual activities of the nettle leaf extract derived CuO and ZnO NPs. However, the nettle leaf extract alone was evaluated for both the properties to identify the efficacy of the extract alone. It can be inferred from the study that the nettle leaf extract derived NPs exhibited augmented antimicrobial and anti-oxidant activities.

Comment 12: P.482 It should be explained why the film was tested for its antibacterial activity against these four pathogens: Campylobacter, E. cloacae, Salmonella, and S. aureus? These bacteria are not spoilage bacteria of guava. What is the significance of the film to delay the spoilage of guava?

Answer: The four human pathogenic bacteria i.e. Campylobacter, E. cloacae, Salmonella, and S. aureus have been used to test the antibacterial potential as these are indicator (S. aureus and Salmonella) or emerging food-borne pathogens (Campylobacter, E. cloacae). These microbes have been used as test organisms to demonstrate the antimicrobial potential of nanocomposite films in several reports (doi:10.4315/0362-028x.jfp-17-509, doi:10.1177/1082013219894202, doi:10.1002/pat.3434). These opportunistic microbes can perpetuate and reside on the surface, and in the internal tissues of various fruits and vegetables. These human pathogens may get inoculated due to use of unprocessed compost or farmyard manure during cultivation, manual picking and handling of the produce, and use of contaminated or unprocessed recycled water.  

The fruit wash water study indicated that the guava fruits packaged in the chitosan nanocomposite films exhibited low mold viable counts compared to unpackaged control fruits. Therefore, these films helped in curbing the spoilage of guava fruit caused by postharvest fungal pathogens. Further, the guava fruit ripening was delayed on nanocomposite film packaging possibly due to oxygen gas barrier properties of the chitosan nanocomposite film which led to decreased respiration and softening of the fruit.      

Comment 13: P.488 For the casting method, it is unlikely that these films appeared to be uniform along all the side. Due to the pulling effect of the surface tension of the solution and the adsorption force between the solution and the mold during the drying process, there are usually wrinkles around the film. And the side of the film in contact with the mold is smoother, and the other side is rougher.

Answer: Yes, the reviewer observations are correct. We have taken the thickness values at ten random spots on the film using the vernier caliper and then obtained the average value. The contents in the manuscript have been revised considering the suggestion of the reviewer.

Comment 14: P.520/P.597-599 It should be confirmed that Fig.5a is SEM or EDS?

Answer: The Figure 5a has four images coded with capital A, B, C and D. The A and B depict the SEM micrographs of the surface of the CuO and ZnO-nanocomposite films. The C and D are the EDS spectra of the same films obtained through an EDS attached to SEM.

Comment 15: P.604 Because the content of CuO and ZnO should be very low (otherwise there will be food safety concerns), the FTIR analysis of the nanocomposite films seems meaningless

Answer: The Figure 5b has been deleted as desired.

Comment 16: P.638 Antioxidant properties of prepared chitosan nanocomposite films was not shown in Fig.4b.

Answer: It was a typing error. The antioxidant activity of the nano-composite films is provided in Fig. 6.

Comment 17: P.679 The author concludes that greater loss in firmness in unwrapped fruit might be due to higher weight loss in the fruit. The firmness of fruit is mainly related to pectin and the structure of fruit matrix, the decrease in water content might lead to higher firmness. Because guava is a climacteric fruit, the authors should discuss the influence of these films on the ripening of guava in order to reasonably explain the physical properties of guava in Figure 6, Figure 7, and Table 3.

Answer: The discussion section has been enhanced elaborating the high respiration rates, and ethylene-induced degradation of the hemicellulose pectin content in guava to be primarily responsible for low shelf life of the unpackaged fruits. The CuO/ZnO NPs chitosan composite films helped in suppression of the respiration rate and pectin degradation by pectin methylesterases. It has been reported that proteins and polysaccharides such as chitosan possess tight network structure due to presence of highly ordered H-bonds which facilitate excellent barrier properties for oxygen gas.

In the control, there was a steep loss in the fruit firmness during storage. This might be due to softening and shriveling of the fruits. The fruit softening is caused by breakdown of insoluble protopectin into soluble pectin and/ or hydrolysis of starch. However, the breakdown of pectic substances in middle lamella of cell wall is the major step in the ripening of fruits resulting in fruit softening. The higher levels of SSC and lower acidity in control fruits also suggest the breakdown of starch and organic acids. The lower acidity levels in the fruit from control might be due to higher respiration rates and higher ripening. Fruit firmness also showed a positive (0.597) correlation with acidity.

Yang L, Paulson AT. Effects of lipids on mechanical and moisture barrier properties of edible gellan film. Food Research International. 2000;33(7):571–578.

Hong SI, Krochta JM. Oxygen barrier performance of whey-protein-coated plastic films as affected by temperature, relative humidity, base film and protein type. Journal of Food Engineering. 2006;77(3):739–745.

Reviewer 2 Report

The manuscript Biomolecules-1070579 reports the preparation and characterization of packaging films based on chitosan embedded with CuO or ZnO nanoparticles stabilized with Urtica dioica leaf extract. CuO and ZnO nanoparticles were obtained by exploiting the reducing activity of biomolecules extracted from Urtica dioicaleaf. The biomolecules have also been thought to provide the packaging with antimicrobial and antioxidant activity. Both the pure nanoparticles and the packaging films were fully characterized.

General comment

The topic of the manuscript is worth of investigation, however data presentation is confused in several parts and the Discussion section is empty. I have also some concerns on the reproducibility of the whole system. In particular, in the packaging preparation, the composition of the leaf extract may provide reproducibility issues. I think the authors should comment on that and provide experimental evidence on the validity of the developed method to provide packaging films with reproducible composition and properties. In addition, the mechanical characterization of the chitosan film, which may be affected by the introduction of CuO and ZnO nanoparticles, has not been addressed by the authors. Finally, a correlation of the chitosan film properties with the amounts of entrapped NPs would have been interesting to investigate.

Specific comments

1) In the introduction, the chitosan antimicrobial activity has been only cited. I suggest to widen a little bit this aspect also in the light of the possibility to improve such activity by chitosan combination with natural extracts (see and cite the following references: https://doi.org/10.1016/j.ejpb.2018.01.012 and https://doi.org/10.1016/j.carbpol.2017.09.073).

2) Page 3 line 102: Please make explicit the acronym GAE

3) Page 3 line 123: Please provide information about chitosan source and molecular weight.

4) Page 4 line 188: How was the amount 25 microliters NPs chosen for testing the antimicrobial activity?

5) Page 5 line 196: How was the amount 25 mg/L NPs chosen for entrapping in the chitosan film?

6) Page 6 line 282: Why was ABTS used for testing the antioxidant activity of the films but not for the pure NPs and extract leaf?

7) The sections 2.2.9.5 and 2.2.9.6 have the same title.

8) Page line 350: Please provide the Cu:O and Zn:O ratio obtained with EDS experiments.

9) Figure 1: SEM micrographs are not very representative and could be deleted. In addition, in the Dynamic Light Scattering graph the legend is missing.

10) Page 9 line 380: Replace C-O bond with C=O.

11) Page 9 line 399: At 3310 cm-1 also O-H stretching absorbs.

12) Page 10 section 3.2: The authors define “remarkable” the antioxidant activity of ZnO and CuO nanoparticles. On which basis? Which is the benchmark? In addition, it would have been useful to study the antioxidant activity of NPs as a function of NPs concentration to determine the EC50 value.

13) Figure 4 and Figure 5: I suggest to revise position of these figures. It is not possible to put the figure in a place of the text and comment it much far ahead. In addition, I suggest to delete Figure 4b and Figure 5b because not significant. Finally, in Figure 5A, it’s not clear to which surfaces the SEM images refer to.

14) Cite Table 2 when you talk of Film Moisture (page 13)

15) Delete section 3.6.2 FTIR spectroscopy of the nanocomposite films. In my opinion, this characterization does not add significant information to the manuscript.

16) Section 3.8.1 Please relate weight loss with Moisture content and film solubility.

17) Table 3 is not cited in the text neither commented.

18) Figure 9 is not cited.

19) The Discussion section has not been written.

Author Response

Reviewer 2:

Comments and Suggestions for Authors

The manuscript Biomolecules-1070579 reports the preparation and characterization of packaging films based on chitosan embedded with CuO or ZnO nanoparticles stabilized with Urtica dioica leaf extract. CuO and ZnO nanoparticles were obtained by exploiting the reducing activity of biomolecules extracted from Urtica dioica leaf. The biomolecules have also been thought to provide the packaging with antimicrobial and antioxidant activity. Both the pure nanoparticles and the packaging films were fully characterized.

 General comment

The topic of the manuscript is worth of investigation, however, data presentation is confused in several parts and the Discussion section is empty. I have also some concerns on the reproducibility of the whole system. In particular, in the packaging preparation, the composition of the leaf extract may provide reproducibility issues. I think the authors should comment on that and provide experimental evidence on the validity of the developed method to provide packaging films with reproducible composition and properties. In addition, the mechanical characterization of the chitosan film, which may be affected by the introduction of CuO and ZnO nanoparticles, has not been addressed by the authors. Finally, a correlation of the chitosan film properties with the amounts of entrapped NPs would have been interesting to investigate.

Reply: We thank the reviewer for acknowledging the volume of the research incorporated in the study. We have improved the contents to omit confusions. The discussion section has been revised substantially. We agree with the comment regarding the reproducibility issue. It is inherent constraint for all green chemical synthesis protocols. However, if careful sample collection for the stinging nettle plants of a specific age growing under particular set of agro-climatic and edaphic conditions will be performed, the obtained extract will contain equivalent amounts of the necessary phytochemicals required for reduction of metal salts to metal oxide NPs. We have performed the total phenol and total flavonoid content estimation of the nettle leaf extract. Further, a LC-MS study can be most appropriate to address variability in the phytochemical content of the leaf extract. We could not perform the characterizations related to mechanical properties of the films due to unavailability of the required facility because of COVID-19 lockdown scenario.   

 Specific comments

Comment 1: In the introduction, the chitosan antimicrobial activity has been only cited. I suggest to widen a little bit this aspect also in the light of the possibility to improve such activity by chitosan combination with natural extracts (see and cite the following references: https://doi.org/10.1016/j.ejpb.2018.01.012 and https://doi.org/10.1016/j.carbpol.2017.09.073).

Answer: The papers discussing the antimicrobial potential of chitosan and natural extracts have been incorporated in the introduction section.

Comment 2: Page 3 line 102: Please make explicit the acronym GAE.

Answer: It is Gallic acid equivalent (GAE). The same has been incorporated at the indicated place in the manuscript.

Comment 3: Page 3 line 123: Please provide information about chitosan source and molecular weight.

Answer: The chitosan was purchased from HiMedia Laboratories Pvt. Ltd., Mumbai, India. The medium molecular weight (190,000 to 300,000 g mol-1) chitosan was used to prepare the films.

Comment 4: Page 4 line 188: How was the amount 25 microliters NPs chosen for testing the antimicrobial activity?

Answer: Both the CuO and ZnO NPs at 10 mg L-1 working concentration were evaluated for antimicrobial activity. As the agar well diffusion assay was performed to evaluate antimicrobial activity, wells of diameter not more than 5 mm were cut. These wells could not accommodate high volumes to be dispensed, therefore, only 25 microliters of the working concentration were poured in the wells to evaluate the antimicrobial effect.  

Comment 5: Page 5 line 196: How was the amount 25 mg/L NPs chosen for entrapping in the chitosan film?

Answer: The amount of 25 mg/L of the NPs was chosen by performing the antimicrobial assay of the prepared films with working concentrations for NPs ranging from 1 to 25 mg/L. The data for the same has not been included in this paper.  

Comment 6: Page 6 line 282: Why was ABTS used for testing the antioxidant activity of the films but not for the pure NPs and extract leaf?

Answer: The ABTS study was performed to check the antioxidant potential of the chitosan nanocomposite films as the DPPH activity for the same was very low compared to the DPPH activity of the CuO/ZnO NPs.

Comment 7: The sections 2.2.9.5 and 2.2.9.6 have the same title.

Answer: Section 2.2.9.6 contains information of spoilage of guava fruits. The same has been corrected in the manuscript.

Comment 8: Page line 350: Please provide the Cu:O and Zn:O ratio obtained with EDS experiments.

Answer: The weight% of the Cu and oxygen was 40.32 and 29.18 respectively (ratio- 1.381). While the Zn and O weight % was 40.14 and 28.71 (ratio- 1.398).

Comment 9: Figure 1: SEM micrographs are not very representative and could be deleted. In addition, in the Dynamic Light Scattering graph the legend is missing.

Answer: The SEM micrographs have been deleted. The legend has been inserted in the DLS graph.

Comment 10: Page 9 line 380: Replace C-O bond with C=O.

Answer: The C-O bond is replaced with C=O.

Comment 11: Page 9 line 399: At 3310 cm-1 also O-H stretching absorbs.

Answer: The indicated information has been incorporated.

Comment 12: Page 10 section 3.2: The authors define “remarkable” the antioxidant activity of ZnO and CuO nanoparticles. On which basis? Which is the benchmark? In addition, it would have been useful to study the antioxidant activity of NPs as a function of NPs concentration to determine the EC50 value.

Answer: The diverse compounds present in the nettle leaf extract were described to possess ‘remarkable’ antioxidant and metal salt reducing activities (line 469 to 470).

The ZnO and CuO NPs were obtained based on the butylated hydroxytoluene (BHT) as standard. We agree to the reviewer’s suggestion regarding use of different NPs concentrations. However, we can incorporate these results provided we get some more time to procure the nettle plants from quite distant place from the Institute.  

Comment 13: Figure 4 and Figure 5: I suggest to revise position of these figures. It is not possible to put the figure in a place of the text and comment it much far ahead. In addition, I suggest to delete Figure 4b and Figure 5b because not significant. Finally, in Figure 5A, it’s not clear to which surfaces the SEM images refer to.

Answer: The Figure 4 has been moved near to the description of the antimicrobial activity. Both 4b and 5b have been deleted as per the suggestion. The descriptions have been inserted in the SEM images of the nanocomposite films in the Figure 5a.  

Comment 14: Cite Table 2 when you talk of Film Moisture (page 13).

Answer: The table 2 has been cited in the section 3.5.3.

Comment 15: Delete section 3.6.2 FTIR spectroscopy of the nanocomposite films. In my opinion, this characterization does not add significant information to the manuscript.

Answer: The section 3.6.2 has been deleted.

Comment 16: Section 3.8.1 Please relate weight loss with Moisture content and film solubility.

Answer: The description regarding weight loss with moisture content and film solubility has been incorporated.

Comment 17: Table 3 is not cited in the text neither commented.

Answer: Table 3 is cited are line number 716 at page 19 and line number 937 at page 25.

Comment 18: Figure 9 is not cited.

Answer: Figure 9 (now Fig 10) has been cited at line number 792 and 794 at page 21.

Comment 19: The Discussion section has not been written.

Answer: The discussion for all the results has been pooled in the Discussion section now.

Reviewer 3 Report

In this manuscript, the authors report the preparation of a chitosan-based nanocomposite film containing Urtica dioica leaf extract derived copper oxide and zinc oxide nanoparticles for shelf-life extension of the packaged guava fruits. This work includes the structural and physicochemical characterization of the nanocomposite films. Also, the studies of the antioxidant and antimicrobial activity were carefully conducted. The paper contains some relevant contributions. The main concerns are reported below.

  1. Page 3, line 124. The average molecular weight is an essential parameter that influences chitosan characteristics and their applications. The authors should include this information in the materials and methods section.
  2. Page 5, line 195. The authors report that “Chitosan (1% w/v) was dissolved in acetic acid (1% v/v) with constant stirring at 40 ºC for 2 hours.”. Typically, chitosan solutions with concentrations of 1 wt% in 1% acetic acid take longer to dissolve (overnight at room temperature). This procedure can affect the transparency of films.
  3. The format of some of the equations can be improved (e.g., equation 3)
  4. Figure 1 should be improved. The graphic (E) has no legend; the inner tables of the graphs (B) are hardly visible. I suggest using the same color for each sample in the different charts (UV, FTIR, DLS).
  5. Page 9, line 421. The pdi obtained for the metal oxide nanoparticles is really high. The authors should explain in the manuscript why this parameter would not affect the application that they want to give the material obtained.
  6. Figure 5 should be improved. The pictures in the chart (a) are hardly visible; the scale on the wavenumber axis on the Figure (b) should be reversed from 4000 to 700 cm-1 (see figure 1); the Figure (c) is hardly visible, maybe
  7. Page 14, line 580. The sentence “… an ideal packaging material must exhibit low water solubility” should be referenced. The term "low" can be subjective and imprecise. Could you interpret that the values obtained from the parameter film solubility are acceptable?
  8. Page 14, line 582. The sentence “… the formation of strong bonds…” is confused. Please explain what kind of particular bonds might take place in this event. Does this correlate with the results obtained by FTIR?
  9. Page 14, lines 612-616. The explanation of "amide 1" band drift is important for discussing the results in this manuscript. Would it be possible to expand the region where this change is observed (Figure 5 (b))?

Author Response

Reviewer 3:

Comments and Suggestions for Authors

In this manuscript, the authors report the preparation of a chitosan-based nanocomposite film containing Urtica dioica leaf extract derived copper oxide and zinc oxide nanoparticles for shelf-life extension of the packaged guava fruits. This work includes the structural and physicochemical characterization of the nanocomposite films. Also, the studies of the antioxidant and antimicrobial activity were carefully conducted. The paper contains some relevant contributions. The main concerns are reported below.

Comment 1: Page 3, line 124. The average molecular weight is an essential parameter that influences chitosan characteristics and their applications. The authors should include this information in the materials and methods section.

Answer: The information has been incorporated in the materials and methods section.

Comment 2: Page 5, line 195. The authors report that “Chitosan (1% w/v) was dissolved in acetic acid (1% v/v) with constant stirring at 40 ºC for 2 hours.”. Typically, chitosan solutions with concentrations of 1 wt% in 1% acetic acid take longer to dissolve (overnight at room temperature). This procedure can affect the transparency of films.

Answer: The higher temperature (40°C) was used to increase the solubility in short time as several combinations of the chitosan films were to be fabricated. The use of higher concentrations of the acetic acid were avoided as this affects the film transparency, tensile strength and elasticity.

Comment 3: The format of some of the equations can be improved (e.g., equation 3).

Answer: The format of equations has been improved. All the equations have been inserted using the ‘insert equation’ function.

Comment 4: Figure 1 should be improved. The graphic (E) has no legend; the inner tables of the graphs (B) are hardly visible. I suggest using the same color for each sample in the different charts (UV, FTIR, DLS).

Answer: The graphs have been improved as per the suggestions.

Comment 5: Page 9, line 421. The pdi obtained for the metal oxide nanoparticles is really high. The authors should explain in the manuscript why this parameter would not affect the application that they want to give the material obtained.

Answer: The pdi for the CuO and ZnO NPs has been high possibly as no synthetic stabilizing agent/ surfactant was utilized during the preparation of the NPs. The DLS and zeta potential studies were not performed for the NPs once synthesis was complete rather the dried NPs were resuspended in deionized water for these studies as the equipment was outsourced and the results were incorporated later. We are now incorporating the revised zeta potential analysis for the fresh samples to avoid the previous discrepancy.   

Comment 6: Figure 5 should be improved. The pictures in the chart (a) are hardly visible; the scale on the wavenumber axis on the Figure (b) should be reversed from 4000 to 700 cm-1 (see figure 1); the Figure (c) is hardly visible, maybe

Answer: The Figure 5b has been removed as per suggestion of the Reviewer 2. The contrast and resolution of the Figure 5a and 5c have been improved for better clarity.

Comment 7: Page 14, line 580. The sentence “… an ideal packaging material must exhibit low water solubility” should be referenced. The term "low" can be subjective and imprecise. Could you interpret that the values obtained from the parameter film solubility are acceptable?

Answer: The film solubility values obtained for the ZnO and CuO-Chitosan films are in good agreement with the values (ranged from 16.49 to 22.36%) obtained by Salari et al (2018) for different chitosan alone, chitosan-nanocellulose and chitosan-AgNPs nanocomposite films.

Salari, M., Sowti Khiabani, M., Rezaei Mokarram, R., Ghanbarzadeh, B., & Samadi Kafil, H. (2018). Development and evaluation of chitosan based active nanocomposite films containing bacterial cellulose nanocrystals and silver nanoparticles. Food Hydrocolloids, 84, 414–423. doi:10.1016/j.foodhyd.2018.05.037

Comment 8: Page 14, line 582. The sentence “… the formation of strong bonds…” is confused. Please explain what kind of particular bonds might take place in this event. Does this correlate with the results obtained by FTIR?

Answer: Yes, it correlates well with the FTIR results as hydrogen bond formation occurs between the amide and glycosidic functional (hydrophilic in nature) with the ZnO NPs. The same has been incorporated in the manuscript.

Comment 9: Page 14, lines 612-616. The explanation of "amide 1" band drift is important for discussing the results in this manuscript. Would it be possible to expand the region where this change is observed (Figure 5 (b))?

Answer: The Figure 5b and the respective contents have been deleted as suggested by the Reviewer 2.

Round 2

Reviewer 2 Report

Responses to some comments are not satisfactory.

Comment 8. the ratios Cu:O and Zn:O ratio obtained with EDS experiments need to be inserted in the text.

Comment 12. The given answer doesn't respond to the comment. Why did the author state the antioxidant activity of ZnO and CuO nanoparticles is remarkable? Remarkable compared to what? Tha found antioxidant actvity is similar or higher than .... which compound has been taken as reference? The authors can also compare their data with similar CuO ZnO nanoparticles present in the literature.

Comment 16. It is not clear where the correlation among weight loss, Moisture content and film solubility was added in the text.

Finally,  the discussion is  in several parts a repetition of the obtained results. 

Author Response

Reviewer 2:

Comments and Suggestions for Authors

Comment 1: Responses to some comments are not satisfactory.

Reply: The replies to the indicated comments have been revised.

Comment 8. the ratios Cu:O and Zn:O ratio obtained with EDS experiments need to be inserted in the text.

Reply: The atom% values of the Cu/Zn and oxygen (O) as obtained through SEM-EDS analysis of the nanoparticles. The atom% values of Cu and oxygen was 16.35 and 46.98 respectively (ratio- 1:2.87). While the Zn and O atom% was 15.56 and 45.74 (ratio- 1:2.93). These ratios indicate occurrence of both peroxide and oxide forms of the copper and zinc NPs. Further, the higher oxygen atom% also indicate towards the contribution of organic compounds derived from the nettle leaf extracts to the oxygen signals in the EDS spectra. Therefore, the oxygen atom% are substantially higher than anticipated to derive the empirical formula for CuO/CuO2 and ZnO/ZnO2 NPs.

Comment 12. The given answer doesn't respond to the comment. Why did the author state the antioxidant activity of ZnO and CuO nanoparticles is remarkable? Remarkable compared to what? Tha found antioxidant actvity is similar or higher than .... which compound has been taken as reference? The authors can also compare their data with similar CuO ZnO nanoparticles present in the literature.

Reply: The antioxidant activity of the ZnO and CuO NPs was compared with the values obtained for the butylated hydroxytoluene (BHT) as standard. The CuO NPs exhibited antioxidant activities similar to BHT standard solution. However, the ZnO NPs possessed antioxidant potential slightly lower than the standard BHT solution.

Similar range of the DPPH radical scavenging activity of CuO NPs derived from fresh leaf extracts of Azadirachta indica, Hibiscus rosasinensis, Murraya koenigii, Moringa oleifera and Tamarindus indica compared to ascorbic acid standard was reported by Rehana et al (2017). Likewise, CuO NPs obtained from Abutilon indicum leaf extract also exhibited DPPH radical scavenging activities equivalent or higher than the BHT standard (Ijaz et al 2017). The DPPH RSA of Allium ursinum leaf extract derived ZnO NPs was found to exhibit a concentration dependence with either similar or slightly higher RSA compared to the standard BHT solution (Rabiee et al 2020). This improved antioxidant activity may be due to the efficient and strong transfer of the d-orbital electrons that helped in stabilization of the DPPH molecule.

 Rehana, D., Mahendiran, D., Kumar, R.S., Rahiman, A.K. (2017). Evaluation of antioxidant and anticancer activity of copper oxide nanoparticles synthesized using medicinally important plant extracts. Biomedicine & Pharmacotherapy, 89, 1067–1077. doi:10.1016/j.biopha.2017.02.101

Ijaz, F., Shahid, S., Khan, S.A., Ahmad, W., Zaman S (2017) Green synthesis of copper oxide nanoparticles using Abutilon indicum leaf extract: Antimicrobial, antioxidant and photocatalytic dye degradation activities. Tropical J. Pharmaceutical Res.16(4): 743-753.

Rabiee, N., Bagherzadeh, M., Kiani, M., Ghadiri, A.M., Zhang, K., Jin, Z., Ramakrishna, S., Shokouhimehr, M. (2020) High gravity-assisted green synthesis of ZnO nanoparticles via Allium ursinum: Conjoining nanochemistry to neuroscience. Nano Express 1:020025. Doi: https://doi.org/10.1088/2632-959X/abac4d

Comment 16. It is not clear where the correlation among weight loss, Moisture content and film solubility was added in the text.

Reply: The relationship among the weight loss (%) of the packaged fruits with the altered gas and water barrier properties of the chitosan nanocomposite films is given in the discussion section (line number 773-776). The correlation of the MC and FS properties of the films has been incorporated in the discussion section (line number 778-783).

Comment: Finally, the discussion is in several parts a repetition of the obtained results. 

Reply: The repetitions in the discussion section have been omitted.

Reviewer 3 Report

The second version of this manuscript is well-organized, and its physicochemical and biological aspects are well described. The work's purpose is more accurate after the suggested corrections, and it deserves to be published.

Author Response

Reviewer 3

Comments and Suggestions for Authors

The second version of this manuscript is well-organized, and its physicochemical and biological aspects are well described. The work's purpose is more accurate after the suggested corrections, and it deserves to be published.

Reply: We thank the reviewer for acknowledging the extensive revision work and approving the contents for publication.